# *C. elegans* orthologs MUT-7/CeWRN-1 of Werner syndrome protein regulate neuronal plasticity

Tsung-Yuan Hsu[1,2,3], Bo Zhang[3], Noelle D L'Etoile[3], Bi-Tzen Juang[1,2]*

[1]Department of Biological Science and Technology, National Yang Ming Chiao Tung University, Hsinchu, Taiwan; [2]Department of Biological Science and Technology, National Chiao Tung University, Hsinchu, Taiwan; [3]Department of Cell and Tissue Biology, University of California, San Francisco, San Francisco, United States

**Abstract** *Caenorhabditis elegans* expresses human Werner syndrome protein (WRN) orthologs as two distinct proteins: MUT-7, with a $3'-5'$ exonuclease domain, and CeWRN-1, with helicase domains. How these domains cooperate remains unclear. Here, we demonstrate the different contributions of MUT-7 and CeWRN-1 to 22G small interfering RNA (siRNA) synthesis and the plasticity of neuronal signaling. MUT-7 acts specifically in the cytoplasm to promote siRNA biogenesis and in the nucleus to associate with CeWRN-1. The import of siRNA by the nuclear Argonaute NRDE-3 promotes the loading of the heterochromatin-binding protein HP1 homolog HPL-2 onto specific loci. This heterochromatin complex represses the gene expression of the guanylyl cyclase ODR-1 to direct olfactory plasticity in *C. elegans.* Our findings suggest that the exonuclease and helicase domains of human WRN may act in concert to promote RNA-dependent loading into a heterochromatin complex, and the failure of this entire process reduces plasticity in postmitotic neurons.

*For correspondence:
btjuang@nctu.edu.tw

Competing interests: The authors declare that no competing interests exist.

## Introduction

Werner syndrome (WS) is an adult-onset progeroid disease in which mutations in the gene encoding the Werner syndrome protein (WRN) are thought to cause abnormal cell function (*Shamanna et al., 2017*). Patients with WS have inactivating mutations in either the $3'-5'$ exonuclease domain or the helicase domain of WRN (*Yu et al., 1996*; *Huang et al., 2006*). Earlier studies of human WS have focused on the relationship between WRN helicase activity and genome integrity, including functions such as DNA repair and telomere maintenance, but the importance of WRN exonuclease activity was recently emphasized according to molecular genetic tests. First, although sequence analysis of individuals with WS shows that ~95% mutations in *WRN* genes produce frameshift and nonsense mutations that are predicted to result in truncated proteins, people harboring mutations causing a 90% reduction in WRN helicase activity but leaving WRN exonuclease activity intact do not present with the clinical manifestations of WS (*Kamath-Loeb et al., 2017*). Second, in 10–15% of patients diagnosed with WS, no mutation is found within WRN (*Oshima and Hisama, 2014*). In some of these non-classical cases, an arginine-to-cysteine substitution is found at amino acid 507 (R507C) in the $3'-5'$ exonuclease domain of POLD, which is a DNA polymerase that associates with the WRN helicase during lagging strand synthesis (*Lessel et al., 2015*). Third, in a *Drosophila melanogaster* model of WS, loss of the Drosophila WRN, which only contains the $3'-5'$ exonuclease domain, affects lifespan under NAD[+] supplementation (*Fang et al., 2019*). However, it is unclear how the exonuclease and helicase domains of the WRN differentially contribute to protection against age-related pathologies. In *Caenorhabditis elegans*, these functions are encoded by two separate nematode proteins,

MUT-7 and CeWRN-1, providing an excellent platform for analyzing how the exonuclease and helicase activities of WRN collaborate to regulate cellular functions.

The nematode *C. elegans* has a highly developed olfactory system and exhibits robust behavioral plasticity upon environmental stimulation. The cells within the sensory circuit include amphid wing cells (termed AWC neurons), which respond to attractive volatile cues such as butanone (*Bargmann et al., 1993*). Prolonged odor exposure in the absence of food causes the animals to ignore a previously attractive odor (*Colbert and Bargmann, 1997*; *L'Etoile et al., 2002*). This behavioral plasticity results from changes within an AWC neuron itself driving a cGMP-dependent protein kinase (PKG), EGL-4, into the nucleus (*Lee et al., 2010a*). A feedback mechanism involved in the process of neuronal plasticity requires a reduction in the mRNA expression levels of a membrane-bound guanylyl cyclase, ODR-1 (*L'Etoile and Bargmann, 2000*). Transcriptional silencing is mediated by loading the heterochromatin-binding protein HPL-2 (an HP1 homolog) onto the *odr-1* locus (*Juang et al., 2013*). The assembly of heterochromatin complexes is directed by an increase in the *odr-1* small interfering RNA (siRNA), consisting of 22 nucleotides starting with a 5′ guanosine (22G), whose synthesis requires the 3′−5′ exonuclease activity of MUT-7 in butanone-trained animals (*Juang et al., 2013*). *C. elegans* MUT-7 shares 29% sequence identity with the 3′−5′ exonuclease domain of human WRN (*Ketting et al., 1999*). In *Arabidopsis thaliana*, loss of the functional *mut-7* ortholog encoding a WS-like exonuclease (WEX) leads to defective post-transcriptional gene silencing (*Glazov et al., 2003*). In *C. elegans*, CeWRN-1 shows 43% sequence identity with the helicase domain of human WRN (*Lee et al., 2004*). Loss of CeWRN-1 seems to cause several progeroid phenotypes, such as decreased lifespan and pharyngeal clogging in the worm head (*Lee et al., 2004*). However, the role of CeWRN-1 in neuronal plasticity has not been investigated. Although the major clinical features of WS do not include significant neurodegenerative disorders, WS patients have recently been observed to show a brain atrophy (*Goto et al., 2013*; *Lebel and Monnat, 2018*). In addition, *Fang et al., 2019* reported WRN-related microarray data in the *C. elegans* brain nervous system, and their findings suggest that WRN may play important roles in neuronal development and neuroplasticity.

We therefore utilized the neuronal plasticity of the *C. elegans* AWC olfactory neuron to elucidate how MUT-7 and CeWRN-1 work together to shape animal olfactory behavior. We report that prolonged odor stimulation results in the production of *odr-1* siRNAs, mediated by the MUT-7 3′−5′ exonuclease in the cytoplasm. These small RNAs, acting as transmitters, facilitate the phosphorylation of MUT-7 and HPL-2 by EGL-4 in the nucleus. The CeWRN-1 helicase associates with the chromatin-binding protein HPL-2 to promote heterochromatin formation to silence ODR-1 expression.

## Results

### MUT-7 and CeWRN-1 mediate behavioral plasticity

The human *WRN* gene encodes a 1432 amino acid protein that possesses an N-terminal 3′−5′ exonuclease domain and three C-terminal helicase domains (*Figure 1A*). Two different nematode proteins are orthologous to the functional domains of human WRN: *C. elegans* MUT-7 contains a 3′−5′ exonuclease and CeWRN-1 has three helicase domains (*Figure 1A*). Our previous studies demonstrated that MUT-7 is also required for generating endogenous 22G siRNAs in response to the prolonged odor stimulation of olfactory AWC neurons (*Juang et al., 2013*). Olfactory behavior was measured by using a well-established chemotaxis assay (*Figure 1B*, upper) in which the animal's sensory neurons are stimulated by a variety of odors, and the neural response is reflected in its behavior. The animal's naïve or primary response is to seek out innately attractive odors (chemotaxis). This odor-seeking response is decreased when the animal experiences odor in the absence of food, causing the animal to ignore the previously attractive odor (olfactory learning). Olfactory behavior is quantified by a chemotaxis index (CI): naïve wild-type animals sense an attractive odor with a high CI value (close to 1.0), while prolonged odor stimulation reduces animal odor-seeking behavior, resulting in a decreased CI (close to 0).

Strains that lacked MUT-7 or CeWRN-1 were analyzed in the chemotaxis assay, and we found that their odor-trained CI was not only greater than half of the naïve CI but also significantly differed from the CIs of odor-trained wild-type animals (*Figure 1B*, lower figure), suggesting that MUT-7 and

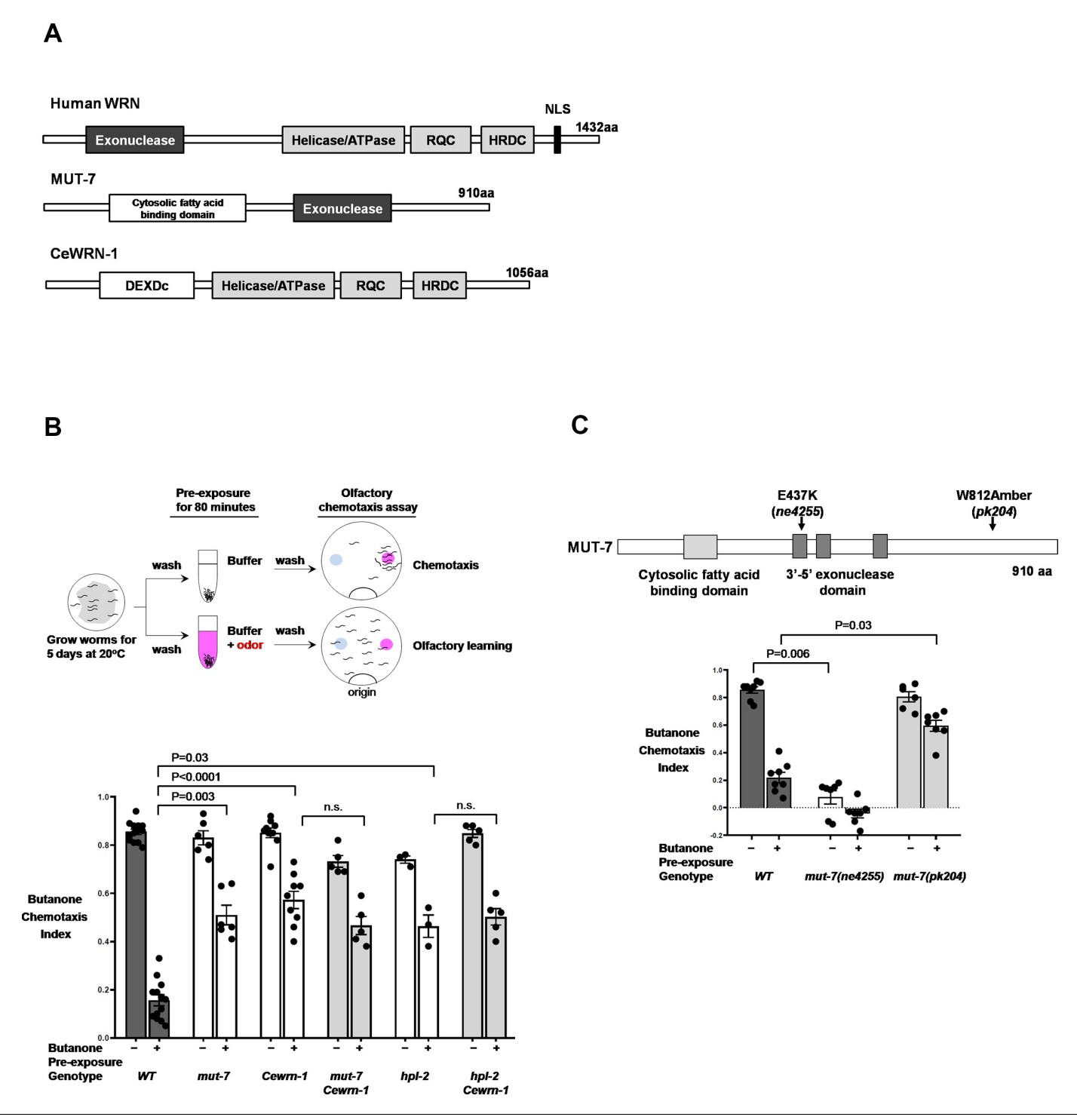

**Figure 1.** The two *C. elegans* orthologs of human WRN required for olfactory learning behavior. (**A**) Alignment of *C. elegans* MUT-7 and CeWRN-1 with human WRN. The 3′−5′ exonuclease domain of nematode MUT-7 has been predicted to show 29% amino acid sequence identity to that of human WRN, as indicated by the dark gray regions. The three helicase domains of human WRN (helicase/ATPase, RecQ C-terminal domain [RQC], and helicase-and-RNaseD C-terminal [HRDC] domains) are conserved in nematode CeWRN-1, as indicated by light gray regions. The helicase/ATPase domain shares 43% identity, and the RQC and HRDC domains share 25% identity (*Lee et al., 2004*). (**B**) MUT-7 and CeWRN-1 function in AWC neurons to promote butanone-related learning at the time of odor exposure. (Upper) Scheme of olfactory learning. Five-day cultured adult animals were washed to remove bacteria, after which half of the population was pre-exposed to buffer alone (top), and the other half was pre-exposed to buffer with a diluted odor, such as butanone (bottom). After 80 min, the animals were placed at the 'origin' of a 9 cm assay plate containing a butanone spot (pink

*Figure 1 continued on next page*

*Figure 1 continued*

circle) and a control ethanol spot (blue circle). The animals allowed roaming around the dish for 2 hr at 20°C, and their olfactory behavior was quantified with the chemotaxis index (CI). (Bottom) The mean CIs are from the number of animals pre-exposed to buffer (−) or diluted odor (+). More than fifty animals were analyzed per assay. We used GraphPad Prism eight software to perform multiple comparisons and p-values from the two-way ANOVA results are presented for the indicated strains. Error bars represent SEM. (C) The enzymatic activity of the MUT-7 3′−5′ exonuclease affects olfactory behavior. (Upper) Schematic diagram of *mut-7* alleles. Two alleles (*ne4255* and *pk204*) are indicated by arrows. (Bottom) The *ne4255* allele results in the loss of exonuclease activity and defective chemotaxis, whereas the *pk204* allele results in low exonuclease activity and the loss of butanone-related learning, while chemotaxis remains normal. Bars represent mean CIs, error bars represent SEM, and p-values represent two-way ANOVA results obtained by using GraphPad software.

CeWRN-1 are required for animals to learn. To examine whether MUT-7 and CeWRN-1 act in the same genetic pathway, we generated *mut-7;Cewrn-1* double mutants and found that the ability of the double-mutant animals to alter their response to butanone was similar to that of single-mutant animals (*Figure 1B*, lower). Thus, the data indicate that MUT-7 and CeWRN-1 act in the same pathway in AWC neurons to promote olfactory learning.

MUT-7 contains a conserved 3′−5′ exonuclease domain, which is predicted to be able to recognize and degrade target mRNAs in the 3′−5′ direction (*Ketting et al., 1999*). Thus, we asked whether an intact 3′−5′ exonuclease domain is required for olfaction. An allele of *mut-7(ne4255)* with a missense mutation at E437K has been predicted to reduce the activity of the 3′−5′ exonuclease by interrupting the predicted Mg$^{2+}$ binding domain (*Gu et al., 2009*). *mut-7 (ne4255)* mutant animals were shown to be defective in butanone chemotaxis (*Figure 1C*). A nonsense allele, *mut-7(pk204),* with the W812 Amber mutation has been proposed to produce a truncated MUT-7 protein that blocks RNAi in the germline (*Ketting et al., 1999*). Worms carrying the W812 Amber mutation in MUT-7 showed reduced 22G RNA levels (*Figure 2—figure supplement 1*), and although they were able to show chemotaxis toward butanone, they were unable to learn to ignore this odor after it was paired with starvation (*Figure 1C*; *Juang et al., 2013*). Although the version of the MUT-7 protein expressed in these animals (*mut-7(pk204)*) contained a complete 3′−5′ exonuclease domain, it failed to silence the transposition of the Tc5 transposon, presumably because of an important role of the intact C-terminus (*Gu et al., 2009*). Thus, we found that the neuroplasticity of AWC neurons requires wild-type MUT-7 activity. Therefore, the *mut-7(pk204)* mutant strain provides an excellent platform for understanding the molecular and cell biological roles of MUT-7 with a 3′-5′ exonuclease domain in promoting learning and memory.

## Roles of nuclear and cytoplasmic MUT-7 in promoting learning

In *C. elegans*, functional studies of MUT-7 have thus far focused on its ability to produce siRNAs in the cytoplasm (*Gu et al., 2009*; *Tops et al., 2005*). We previously found that the expression of MUT-7 with GFP appended to its N-terminus (GFP-MUT-7 in *Figure 2A*, upper) specifically in AWC cells restored odor learning in *mut-7(pk204)* mutant animals (*Figure 2B*, fourth pair). The expression of GFP-MUT-7 in AWC also restored *odr-1* 22G RNA levels in *mut-7(pk204)* odor-trained worms (*Figure 2C*, third dataset) (*Juang et al., 2013*). Indeed, we used quantitative real-time PCR to probe *odr-1* mRNA and found that prolonged odor exposure decreased *odr-1* mRNA levels in wild-type animals, while *odr-1* mRNA levels were insensitive to odor exposure in *mut-7(pk204)* mutant animals (*Figure 2—figure supplement 2*). Furthermore, we used a CRISPR-Cas9 system to generate an integrated line expressing ODR-1::GFP under the control of the endogenous *odr-1* promoter. GFP-tagged ODR-1 was concentrated in the flattened ciliated end of the AWC neuron (*Figure 2D*). The fluorescence intensity in naïve wild-type animals was significantly brighter than the fluorescence of GFP in odor-trained wild-type animals (*Figure 2D*, top). In contrast, no significant difference in GFP expression induced by odor exposure was observed in the *mut-7(pk204)* mutants (*Figure 2D*, bottom). These results confirm that MUT-7 is required in the synthesis of *odr-1* 22G RNA after prolonged odor treatment to specifically downregulate the expression of both the *odr-1* mRNA and the ODR-1 protein.

Although the rescuing (active) form of MUT-7 was found in both the nucleus and cytoplasm (*Figure 2A*, top) (*Juang et al., 2013*), we were able to restrict MUT-7 to the cytoplasm or nucleus. By appending GFP to the C-terminus of MUT-7 (called CterGFP-MUT-7, *Figure 2A*, middle), MUT-7 was restricted to the cytoplasm, and by appending four nuclear localization sequences to mCherry

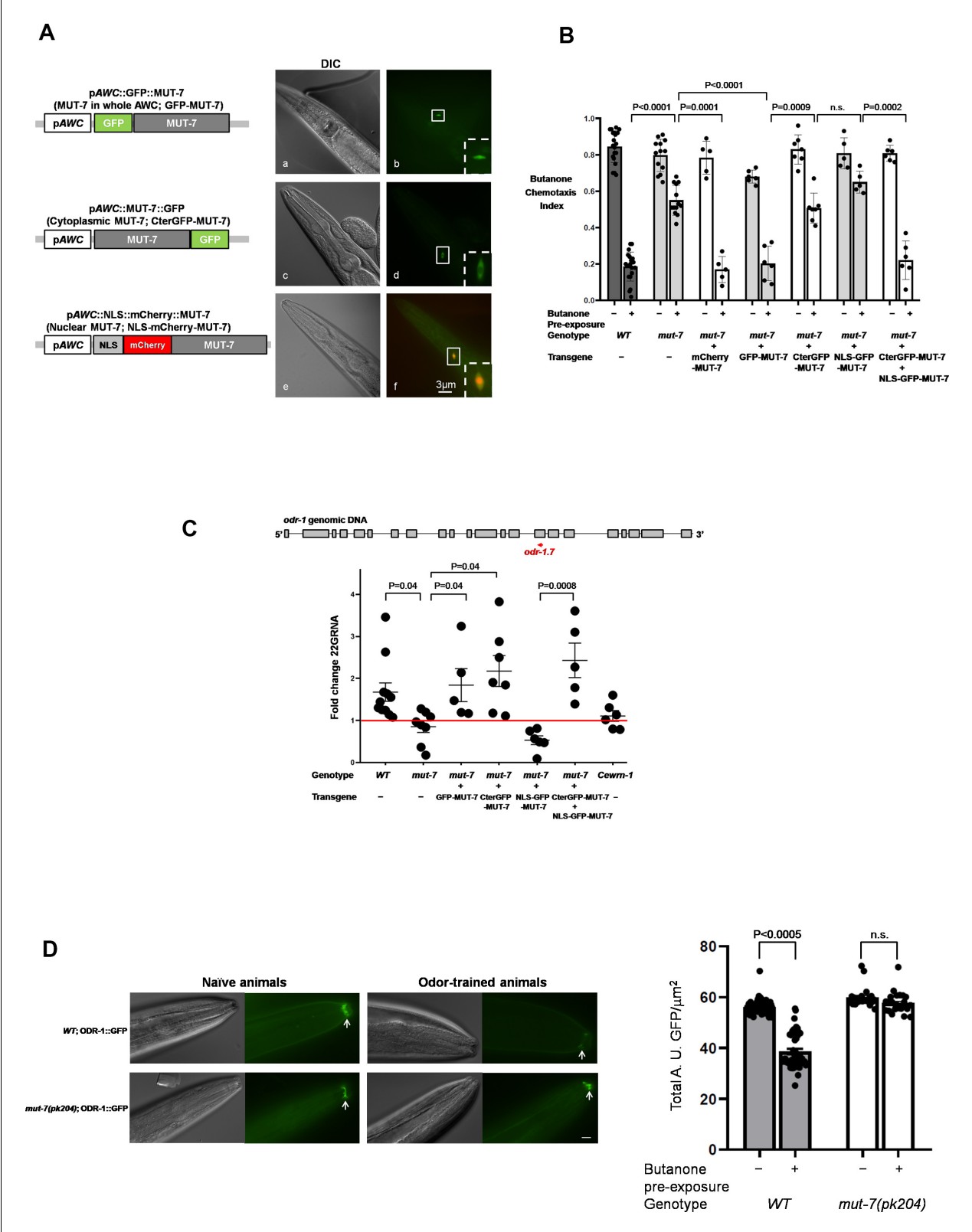

**Figure 2.** Different intracellular roles of MUT-7 mediate olfactory learning. (**A**) Different localizations of MUT-7 in AWCs associated with different positions of the GFP tag. MUT-7 with an upstream (**a and b**) or downstream (**c and d**) GFP tag showed localization throughout the soma (**b**) or in the cytoplasm (**d**), respectively. A 4XNLS fragment was added upstream of N-terminal mCherry-tagged MUT-7 to cause nuclear accumulation in AWCs (**e** and **f**; the background green cytoplasmic signal comes from the coinjection marker p*AWC*::GFP). All constructs were expressed under an AWC-specific

*Figure 2 continued on next page*

*Figure 2 continued*

promoter (p*AWC*). (**B**) MUT-7 localization in AWCs affects olfactory behaviors. The individual constructs from (**A**) were introduced into *mut-7(pk204)* mutants, and the olfactory behavior of the transgenic animals was tested after 80 min of pre-exposure to either buffer alone (−) or diluted butanone (+). All strains analyzed in (**B**) and (**C**) were integrated lines obtained by using UV/TMP methods. The p-values come from two-way ANOVA results obtained by comparing the indicated odor-trained populations. n.s. indicates no significant difference. (**C**) Cytoplasmic MUT-7 is required for the synthesis of *odr-1* 22G RNAs after prolonged odor stimulation. (Upper) Total RNA was extracted from whole animals and *odr-1* 22G RNA was quantified by RT-qPCR with an *odr-1.7* TaqMan probe. (Bottom) The expression of *odr-1* siRNA was normalized to that of odor-insensitive sn2343 RNA, and the fold change between odor-trained and naïve animals of the indicated genotypes was then calculated. The red line indicates no change between odor-trained and naïve populations. The p-values displayed come from the comparison of the fold change between the indicated strains by using one-way ANOVA. (**D**) Prolonged odor exposure decreases endogenous ODR-1 expression. (Left) A gene encoding ODR-1::GFP under the control of an endogenous promoter was integrated into the worm genome by using a CRISPR-based method. GFP was observed in the flattened ciliated end of the AWC neuron indicated by the white arrows. All images were captured using an upright microscope (Leica DM6B) at 63X magnification. In odor-trained wild-type animals, ODR-1 expression decreased the fluorescence intensity by 30% compared to that in naïve wild-type animals. The fluorescence intensity in *mut-7(pk204)* mutant animals was not significantly different between the naïve and odor-trained populations. The fluorescence intensity in the naïve and odor-trained animals was quantified as shown in the right panel. The p-value comes from the comparison of fluorescence intensity between naïve and odor-trained worms by using two-way ANOVA. Error bars represent SEM, and n.s. indicates no significant difference.

The online version of this article includes the following figure supplement(s) for figure 2:

**Figure supplement 1.** Pairwise comparison of *odr-1* 22G RNA levels.
**Figure supplement 2.** Prolonged odor exposure decreases *odr-1* mRNA expression.

---

(termed NLS-mCherry-MUT-7, *Figure 2A*, bottom), MUT-7 was limited to the nucleus. In addition, we tested whether mCherry-MUT-7 (no NLS) was a rescuing form of MUT-7 and found that the expression of mCherry-MUT-7 rescued the learning defects of *mut-7(pk204)* mutant animals (*Figure 2B*, third pair). Furthermore, NLS-mCherry-MUT-7 and NLS-GFP-MUT-7 constructs were generated in parallel, but the NLS-GFP-MUT-7 strain was the only integrated line to be obtained by using a standard UV/trimethylpsoralen (UV/TMP) integration method. Therefore, the NLS-GFP-MUT-7 expression line was used in the following experiments.

To assess how MUT-7 localization affects siRNA levels and how this in turn impacts odor learning, we asked whether the subcellular localization of MUT-7 affected *odr-1* 22G RNA levels and behavioral plasticity in odor-trained animals. We found that animals in which MUT-7 was restricted to the cytoplasm (CterGFP-MUT-7) showed an increase in the *odr-1.7* 22G RNA levels to the same level found in the wild type when they were odor trained (*Figure 2C*, first versus fourth datasets). This was interesting because although they exhibited wild-type levels of *odr-1.7* 22G RNA, these animals were not able to learn as well as the wild types (*Figure 2B*, first versus fifth pairs of bars). By contrast, restricting MUT-7 to the nucleus (NLS-GFP-MUT-7) blocked the odor-dependent increases in both 22G RNA (*Figure 2C*, first versus fifth dataset) and learning (*Figure 2B*, first versus sixth pairs of bars). Importantly, each version of MUT-7 was functional, as expressing both CterGFP-MUT-7 and NLS-GFP-MUT-7 in the *mut-7(pk204)* mutants restored both 22G RNA levels and odor learning (*Figure 2B*, seventh pair of bars, and *Figure 2C*, sixth dataset). Thus, 22G RNA production is not sufficient to cause odor learning, and MUT-7 must perform an additional role in the nucleus, because when it is restricted from the nucleus, animals do not learn.

Taken together, our behavioral and RT-qPCR results indicate that cytoplasmic MUT-7 may function in the small RNA synthesis process. These results also provide insight that MUT-7 acts in a different way in the nucleus. Moreover, the two pools of MUT-7 must (indirectly) act in coordination to promote olfactory learning.

Olfactory learning is assessed in the adult stage, but developmental defects such as an abnormal cell fate could indirectly affect this process. The two AWC neurons are asymmetric with respect to the odors to which they respond. This asymmetry is determined by the expression of STR-2 in either the left or right AWC neuron in wild-type animals (referred to as the AWC[ON] neuron) (*Troemel et al., 1999*). To determine whether the various versions of MUT-7 affect this cell fate, STR-2-driven DsRed fluorescence was expressed in CterGFP-MUT-7 or NLS-GFP-MUT-7 transgenic animals, and asymmetric expression of STR-2 was observed in more than 96% of tested animals (*Supplementary file 1*). This result suggests that the impairment of behavioral plasticity in adult animals carrying CterGFP-MUT-7 or NLS-GFP-MUT-7 is not due to changes in cell fate.

## MUT-7 and EGL-4 interact in the nucleus of odor-trained animals in a PKG phosphorylation site-dependent manner

What is the role of nuclear MUT-7 in determining the chromatin changes seen in odor-trained animals? Previous data have shown that MUT-7 is required not only in the cytoplasm of odor-trained animals, to increase *odr-1* 22G RNA levels (*Figure 2C*), but also in the nucleus, to promote learning (*Figure 2B*). Thus, we wanted to understand the potential function of MUT-7 in the nucleus to promote odor learning. Our prior studies indicated that, for odor learning to occur, PKG EGL-4 needs to enter the nucleus and presumably phosphorylate its targets. Mutations in the PKG consensus sites of MUT-7 caused defects in odor learning (*Figure 3A*; *Juang et al., 2013*). Our previous genetic epistasis experiments indicated that MUT-7 acts downstream of nuclear EGL-4 (*Juang et al., 2013*). Thus, MUT-7 might be phosphorylated by EGL-4 in the AWC nucleus of an odor-trained animal.

Since the expression of full-length MUT-7 in bacteria for in vitro kinase assays was not successful, we next attempted to detect a physical interaction between EGL-4 and MUT-7 by using in vivo bimolecular fluorescence complementation (BiFC). The basic principle of the BiFC assay is the reconstitution of a fluorescent molecule such as the GFP-derived Venus fluorophore that has been split into two separate halves (*Hu et al., 2006*; *Kerppola, 2006*; *Shyu et al., 2008*). Each half of the molecule is appended to a distinct protein, and if the two proteins physically interact, the N- and C-halves of the split Venus protein will reconstitute the fluorophore.

We expressed MUT-7 and EGL-4 tagged with the N- and C-terminal fragments of the Venus fluorophore (VN and VC), respectively, in the AWC neurons of a wild-type animal (*Figure 3B*). We observed that only 1% of buffer-trained animals showed a reconstituted fluorescent signal in the AWC nucleus and that this percentage rose to 75% after prolonged odor training (*Figure 3B and C*, first row). The BiFC signal is binary either on or off. Thus, in a population of naïve worms, few show the signal, while in a population of butanone exposed ones, many showed the signal. This means that individual worms may have complexes between these proteins in the nucleus but the proportion of the population with these complexes changes with butanone treatment. This may reflect the fact that most but not all naive animals are attracted to butanone, but the proportion that is attracted decreases when worms are starved in the presence of butanone. Importantly, the BiFC signal in each butanone-trained animal was seen in only one AWC neuron. This reflects the fact that butanone is sensed by only one AWC (*Wes and Bargmann, 2001*). These results also indicate that our BiFC system can specifically detect protein interactions in the single butanone-sensing neuron of the worm. In addition, we tested constitutively nuclear EGL-4 tagged with the Venus C-terminus (NLS::EGL-4) in naïve animals expressing MUT-7 tagged with the Venus N-terminus, and 86% of odor-trained animals exhibited the BiFC signal (*Figure 3C*, third row). These results indicate that EGL-4 associates either directly or in a protein complex with MUT-7 in the AWC nucleus of odor-trained worms.

We next asked whether the consensus PKG phosphorylation sites in MUT-7 are important for this association between MUT-7 and EGL-4. Thus, we performed a similar BiFC analysis with MUT-7 variants in which all seven of the predicted PKG phosphorylation sites were mutated to nonphosphorylatable alanine residues (*Figure 3A*). These nonphosphorylatable forms of MUT-7 failed to reconstitute the BiFC signal (*Figure 3C*, second and fourth rows). The single S883A mutation (at a site deleted in the pk204 allele of mut-7) was sufficient to block BiFC fluorescence (*Figure 3C*, sixth row). Taken together, these data lead us to propose that the phosphorylated form of MUT-7 is bound by EGL-4 in the AWC nucleus and that EGL-4 kinase may be responsible for phosphorylating MUT-7. As a control, a point mutation in MUT-7 (S516A) that does not affect learning (*Figure 3A*, fifth pair for behavior) (*Juang et al., 2013*) was shown to not disrupt BiFC between EGL-4 and MUT-7 (*Figure 3C*, fifth row).

## The import of *odr-1* 22G RNA by NRDE-3 bolsters MUT-7 and EGL-4 association in the nucleus

We next asked whether 22G RNA production is required for EGL-4 and MUT-7 to associate in the nucleus. The rationale for this question was that small RNA species may direct repressive chromatin marks to specific genes in the AWC nucleus and that EGL-4 creates a repressed chromatin state by phosphorylating HPL-2 (*Juang et al., 2013*). Thus, we asked whether small RNA levels affect the association between EGL-4 and MUT-7 in the nucleus upon odor exposure. To test this hypothesis, we decided to block 22G RNA production by creating a dominant-negative cytoplasmic MUT-7 and

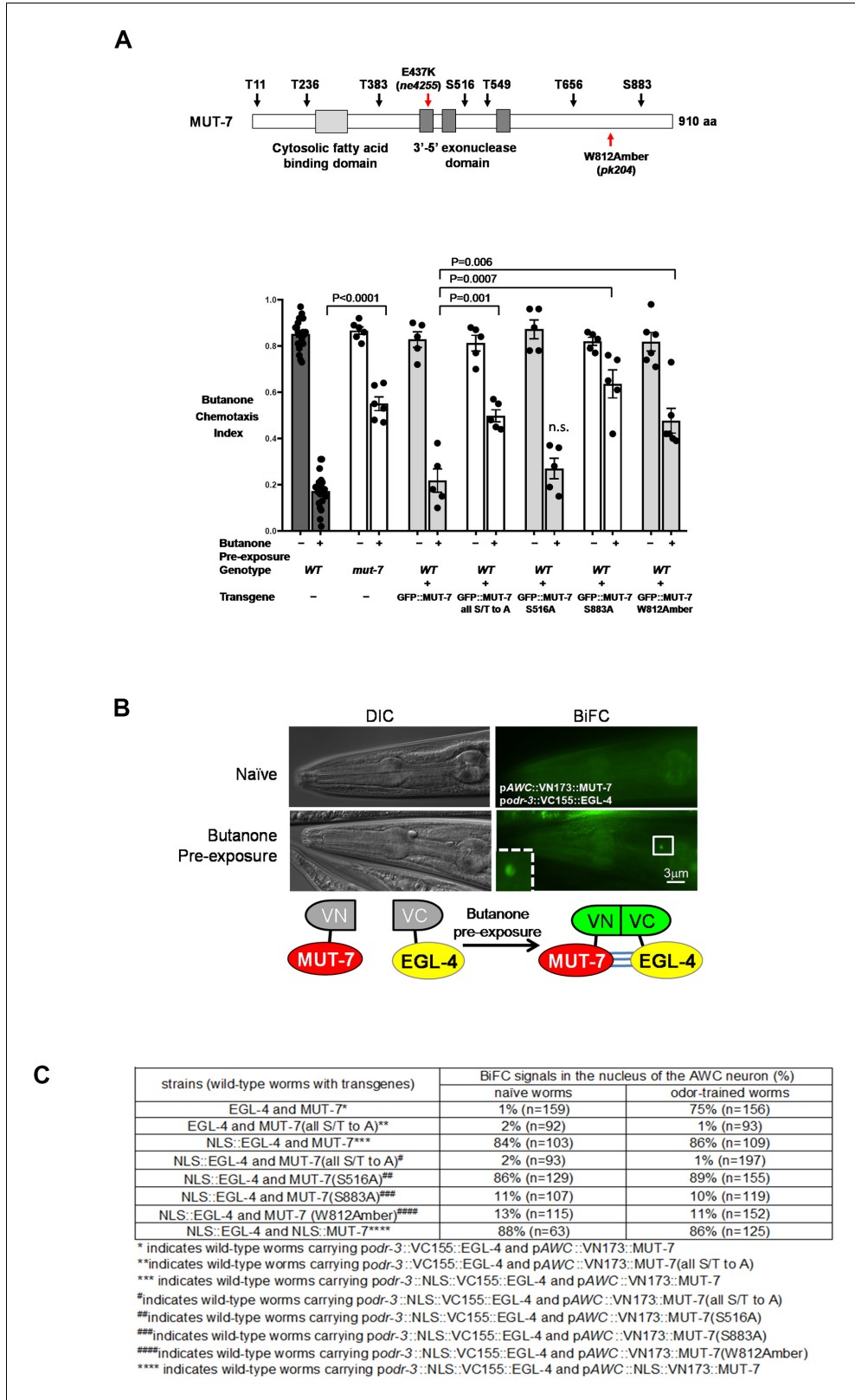

**Figure 3.** MUT-7 is phosphorylated by EGL-4 in the nucleus after prolonged odor exposure. (A) Schematic representation of the seven predicted phosphorylation sites of PKG in MUT-7 in the top panel. Different point mutations were introduced by site-directed mutagenesis, and the constructs were transferred to wild-type animals to generate dominant-negative strains for behavioral assays. The p-value comes from the results of two-way ANOVA for the comparison of the indicated strains. n.s. indicates no significant difference between the wild type and the indicated mutant animals. (B)
*Figure 3 continued on next page*

*Figure 3 continued*

The BiFC assay revealed an in vivo interaction between EGL-4 and MUT-7 in the AWC nucleus. (Upper) BiFC florescence signals were observed in odor-trained worms (bottom) but not in naïve worms (top). (Bottom) Schematic representation of the BiFC constructs. (C) The BiFC screen showed the critical residues within MUT-7 for the specific interaction with nuclear EGL-4. The BiFC screen revealed that the specific interaction between nuclear MUT-7 and EGL-4 was consistent with the adaptation results shown in (B). Since nuclear EGL-4 induced adaptation of the odor-seeking behavioral response in naïve worms, the percentage of BiFC signals in naïve worms expressing NLS::EGL-4 with different versions of MUT-7 is similar to that in the odor-trained worms.

asked if the nuclear pool of MUT-7 would still be able to associate with EGL-4. To test this hypothesis, tryptophan 812 was replaced with a nonsense codon to mimic the sequence variation in *mut-7 (pk204)*. The expression of this construct in wild-type worms resulted in not only failure to learn (*Figure 3A*, seventh pairs) but also failure to produce *odr-1* 22G RNA after odor training (*Figure 2C*). We next asked whether MUT-7(W812Amber) would associate with nuclear EGL-4 in this new context without 22G RNA. We found that only 11% of odor-trained transgenic animals showed the BiFC signal (*Figure 3C*, seventh row). This result suggests that the production of *odr-1* 22G RNA by processes that involve MUT-7 in the cytoplasm is required for the association between EGL-4 and MUT-7 in the AWC nucleus.

We have presented evidence that the cytoplasmic and nuclear pools of MUT-7 do not have to move between the compartments to promote odor learning (*Figure 2B*; the co-expression of NLS-GFP-MUT-7 and CterGFP-MUT-7 in *mut-7(pk204)* animals rescues *mut-7* learning defects). Therefore, we wondered how the 22G RNA-based signal might travel from the cytoplasm to the nucleus. One candidate potentially mediating this process is the Argonaut NRDE-3, which shuttles 22G siRNA from the cytoplasm to the nucleus in the nuclear RNA silencing pathway (*Guang et al., 2008*). A previous coimmunoprecipitation analysis revealed that *odr-1* 22G RNAs are loaded onto NRDE-3 in butanone odor-trained worms (*Juang et al., 2013*), indicating that this protein is a good candidate for the conduit between the cytoplasm and nucleus. To understand whether the entry of *odr-1* 22G RNAs into the nucleus is required to trigger the association between MUT-7 and EGL-4, we turned to BiFC. We expressed MUT-7 tagged with the Venus N-terminus and EGL-4 tagged with the Venus C-terminus in *nrde-3(gg66)* mutant animals lacking NRDE-3. We found that the BiFC signal was not detected in either naïve or odor-trained transgenic worms (*Figure 4A* and *Figure 4—figure supplement 1*, first row). Furthermore, we observed that 86% of odor-trained wild-type worms co-expressing NLS::EGL-4 and constitutively nuclear MUT-7 tagged with the Venus N-terminus (NLS::MUT-7) showed a BiFC signal (*Figure 3C*, last row) but that only 5% of odor-trained *nrde-3(gg66)* mutant worms expressing the same constructs showed a BiFC signal (*Figure 4D*, first row). Taken together, these observations indicate that the import of 22G RNAs into the nucleus via NRDE-3 is required for the association between nuclear MUT-7 and EGL-4.

## MUT-7 associates with CeWRN-1 in the nucleus to promote olfactory learning

WS patients accelerate aging after puberty; thus, mutations in the helicase domains of the human WRN protein have been studied as a possible way to understand the aging process. The WS helicase is orthologous to *C. elegans* CeWRN-1 (*Figure 1A*). CeWRN-1 is involved in multiple cellular events, including DNA replication and repair (*Lee et al., 2010b*).

Olfactory behavioral analysis revealed that strains lacking CeWRN-1 were unable to learn to ignore butanone after training (*Figure 1B*). The expression pattern of CeWRN-1 has not been characterized in vivo, so we placed GFP expression under the control of the *Cewrn-1* promoter and observed a GFP signal in many cells, including the pair of AWC neurons (*Figure 4—figure supplement 2*). To determine whether CeWRN-1 acts in AWC neurons to regulate odor sensing and learning, the CeWRN-1 protein was expressed in AWCs and rescued the learning defects of the *Cewrn-1 (gk99)* mutant (*Figure 4B*, bottom). Furthermore, GFP-tagged CeWRN-1 stained the AWC nucleus (*Figure 4B*, top). These data indicate that CeWRN-1 acts within AWC neurons to promote olfactory plasticity. This is unlikely to be due to a change in AWC cell fate because the wild-type pattern of asymmetric STR-2 expression was observed in all tested animals (*Supplementary file 1*).

The nuclear accumulation of EGL-4 in odor-trained wild-type animals is required for olfactory learning and adaptation (*L'Etoile et al., 2002*; *O'Halloran et al., 2009*; *O'Halloran et al., 2012*;

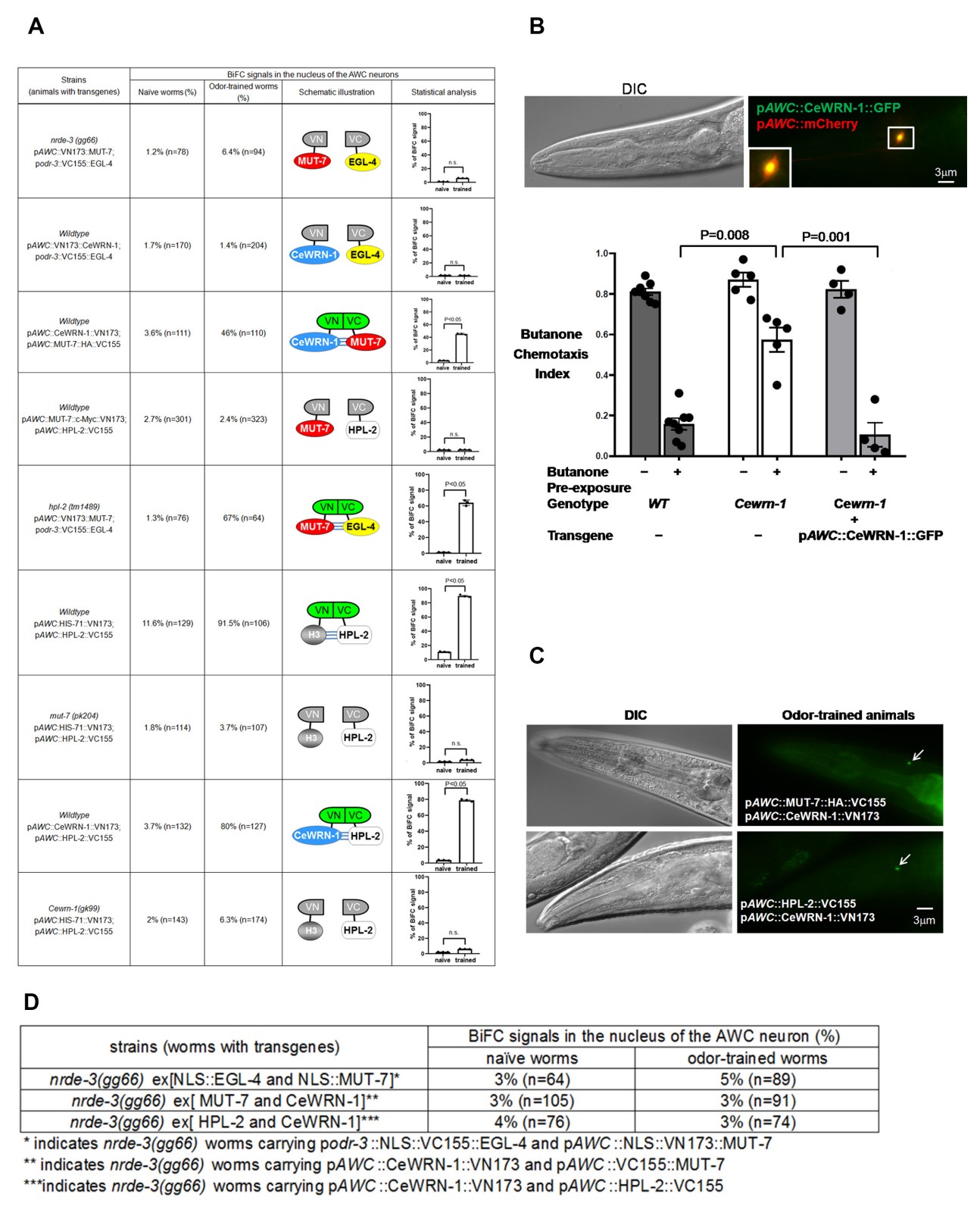

**Figure 4.** CeWRN-1 function in the AWC nucleus is required for olfactory learning. (**A**) The BiFC screen showed in vivo specific protein interactions in the AWC nucleus after prolonged odor exposure. The BiFC fluorescent signals were scored in naïve and odor-trained worms. Each strain was examined in three separate experiments, and all the data were statistically analyzed by two-way ANOVA. (**B**) (Upper) CeWRN-1 is expressed in the AWC nucleus. Fluorescent images of wild-type animals expressing *Cewrn-1* cDNA with a GFP tag at the C-terminus showed protein accumulation in the nucleus.
*Figure 4 continued on next page*

**Figure 4 continued**

(Bottom) The expression of CeWRN-1 in AWC rescues the learning defects of *Cewrn-1(gk99)* mutant animals. GFP-tagged CeWRN-1 was introduced into the *Cewrn-1(gk99)* null mutant and restored olfactory learning ability. The p-values come from two-way ANOVAs between the specified groups. (C) CeWRN-1 interacts with MUT-7 and HPL-2. An influenza hemagglutinin (HA) peptide was inserted between MUT-7 and GFP, and fluorescent images showed that MUT-7 was distributed throughout the AWC cell body, as shown in *Figure 4—figure supplement 3*. Thus, the same strategy was applied to generate the p*AWC*::MUT-7::HA::VC155 construct (top) in the BiFC assay. (D) NRDE-3 is required for the MUT-7 and EGL-4 interaction, the CeWRN-1 and HPL-2 interaction, and the MUT-7 and CeWRN-1 interaction in the AWC nucleus. We expressed the indicated BiFC constructs in *nrde-3(gg66)* mutants and scored the BiFC signals in naïve and odor-trained worms.

The online version of this article includes the following figure supplement(s) for figure 4:

**Figure supplement 1.** Images of BiFC screening results.
**Figure supplement 2.** Localization of CeWRN-1 in AWC neurons.
**Figure supplement 3.** Expression of MUT-7::HA::GFP in the whole cell body of the AWC neuron.

*Cho et al., 2016*); therefore, we asked whether CeWRN-1 is required for EGL-4 nuclear entry. We found that GFP-tagged EGL-4 entered the nucleus of AWC neurons in *Cewrn-1(gk99)* animals that had been trained with butanone (*Supplementary file 2*). Thus, CeWRN-1 does not regulate EGL-4 nuclear entry. We next asked whether CeWRN-1 might interact with EGL-4 in the nucleus. To test this hypothesis, we appended the N-terminal half of Venus to CeWRN-1 and expressed this construct in a strain-expressing EGL-4 tagged with the C-terminal half of Venus. Less than 2% of transgenic animals under either naïve or trained conditions showed a fluorescent signal (*Figure 4A* and *Figure 4—figure supplement 1*, second row), suggesting that the CeWRN-1 and EGL-4 do not interact directly.

MUT-7 and CeWRN-1 are predicted to be *C. elegans* orthologs of the 3′−5′ exonuclease and helicase domains of human WRN, respectively. To explore whether nematode MUT-7 and CeWRN-1 could interact, the BiFC constructs for CeWRN-1 and MUT-7 were co-expressed in wild-type animals, and 46% of odor-trained worms showed the BiFC signal, in contrast to <4% of buffer-trained control worms (*Figure 4A* and *Figure 4—figure supplement 1*, third row; *Figure 4C*, top). In addition, the same BiFC constructs were expressed in *nrde-3(gg99)* mutants, and only 3% of naïve and odor-rained worms exhibited the BiFC signal (*Figure 4D*, second row). These results indicate that CeWRN-1 may associate with nuclear MUT-7 in the presence of NRDE-3 and that they may mediate olfactory learning together.

## Odor-activated MUT-7 directs HPL-2 loading on histone H3

Our previous survey of genes required for olfactory plasticity assumed that MUT-7 may interact with HPL-2 at the time of odor exposure to promote butanone-related learning (*Juang et al., 2013*). Moreover, chromatin immunoprecipitation studies showed that odor adaptation resulted in the loading of HPL-2 onto the *odr-1* locus within AWC neurons (*Juang et al., 2013*). However, it remained unclear whether MUT-7 and HPL-2 act together in a complex or in series in a process promoting HPL-2 loading onto target genes. To distinguish between these possibilities, we first asked whether MUT-7 and HPL-2 are in close enough association in the nucleus to reconstitute fluorescence (BiFC). We found that very few naïve or odor-trained animals showed BiFC (*Figure 4A* and *Figure 4—figure supplement 1*, fourth row). Thus, MUT-7 and HPL-2 are unlikely to be associated even after odor training.

Second, we asked if the EGL-4-MUT-7 association requires HPL-2. We found that, in *hpl-2* null animals, 67% of odor-trained animals showed BiFC, in contrast to only 1% of naïve animals (*Figure 4A* and *Figure 4—figure supplement 1*, fifth row). Thus, neither does HPL-2 exist in a complex with MUT-7 nor is it required to promote the association between MUT-7 and EGL-4.

Next, we asked whether the association of HPL-2 and the histone H3.3 variant HIS-71, which is incorporated into the nuclei of almost all somatic cells of *C. elegans* throughout its lifespan (*Ooi et al., 2006*), increases during odor training, as predicted by the chromatin IP (*Juang et al., 2013*). When we expressed HIS-71 tagged with the Venus N-terminus and HPL-2 tagged with the Venus C-terminus in wild-type animals, we found that 91.5% of odor-trained worms showed BiFC, in contrast to a background rate of 11.6% in naïve worms (*Figure 4A* and *Figure 4—figure supplement 1*, sixth row). Thus, odor training increases the association between HPL-2 and H3.3. To determine whether MUT-7 is required for the increased association, the same set of BiFC constructs was

expressed in the *mut-7(pk204)* mutant background, and less than 4% of either naïve or odor-trained animals exhibited the BiFC signal (*Figure 4A* and *Figure 4—figure supplement 1*, seventh row). This suggests that MUT-7 is required for the formation of the HPL-2-H3 complex in odor-trained animals but that it does not directly associate with either HPL-2 or histone H3.3. This led us to hypothesize that phosphorylated MUT-7 directs the heterochromatin complex to genetic loci, possibly using 22G RNA as a guide.

## CeWRN-1 guides the HPL-2 and histone H3 association

To better understand whether *C. elegans* CeWRN-1 is involved in heterochromatin formation by binding to HPL-2, we first asked whether CeWRN-1 and HPL-2 act in the same genetic pathway. We generated *hpl-2;Cewrn-1* double mutants and found that the odor learning ability of the double mutants was similar to the odor learning ability of *Cewrn-1* or *hpl-2* single-mutant animals (*Figure 1B*, lower). The results indicate that CeWRN-1 and HPL-2 act in the same pathway to promote learning. Next, we asked whether CeWRN-1 associates with HPL-2 to regulate signaling. We co-expressed CeWRN-1 tagged with the N-terminal half of Venus and HPL-2 tagged with the C-terminal half of Venus in wild-type worms. We found that 80% of odor-trained worms but less than 4% of naïve worms showed reconstituted fluorescence in the AWC nucleus (*Figure 4A* and *Figure 4—figure supplement 1*, eighth row; *Figure 4C*, bottom). The same BiFC constructs were expressed in *nrde-3(gg66)*, and less than 4% of naïve or odor-trained worms exhibited the BiFC signal (*Figure 4D*, third row). Therefore, these results indicate that the association between HPL-2 and CeWRN-1 in odor-trained animals depends on NRDE-3. To determine whether CeWRN-1 affects HPL-2 binding to histone H3, we examined the BiFC of HIS-71 and HPL-2 in *Cewrn-1(gk99)* null mutants. We found that loss of CeWRN-1 reduced the BiFC signal, as <7% of naïve and odor-trained worms showed fluorescence (*Figure 4A* and *Figure 4—figure supplement 1*, ninth row). These results showed that CeWRN-1 is required for HPL-2 to associate with histone H3 in odor-trained animals. Our data were also consistent with the observation that the human WRN protein associates with the heterochromatin-binding protein HP1 and histone H3K9me3 in a human WS mesenchymal stem cell model (*Zhang et al., 2015*).

If CeWRN-1 is required for heterochromatin formation, then it is possible that *odr-1* 22G RNA production could be affected by loss of CeWRN-1. Specifically, since odor learning requires the downregulation of ODR-1 by *odr-1* 22G RNAs, we asked whether *odr-1* 22G RNA production is interrupted by loss of CeWRN-1. To test the hypothesis that CeWRN-1 drives 22G RNA function, we asked whether the increase in the expression of *odr-1* 22G RNAs during odor training requires CeWRN-1. We found that in *Cewrn-1* loss-of-function mutants, *odr-1* 22G RNA levels did not change (*Figure 2C*, seventh data set). Importantly, *odr-1* 22G RNA expression at baseline was the same in wild-type worms and *Cewrn-1* mutants (*Figure 2—figure supplement 1*, left and right). By contrast, in *mut-7(pk204)* mutants, the *odr-1* 22G RNA level was significantly decreased in odor-trained animals (*Figure 2—figure supplement 1*, middle). These results suggest that endogenous MUT-7 functions normally in the *Cewrn-1* mutant background. Taken together, the data imply that CeWRN-1 is required for HPL-2 to load onto histone H3 and thus fine tune gene expression via the small RNA-mediated silencing pathway in AWCs during the butanone-related learning process.

## Discussion

We report that the exonuclease and helicase domains of the *C. elegans* orthologs of the WS protein play different roles in siRNA synthesis (mediated by cytoplasmic MUT-7) and heterochromatin formation (mediated by nuclear CeWRN-1) to promote neuronal plasticity. Although there is no obvious clinical outcome of human WS associated with brain diseases, the brain atrophy observed in WS patients (*Goto et al., 2013*; *Lebel and Monnat, 2018*) and the abnormal neuronal gene expression indicated by microarray data from the *C. elegans* WS model (*Fang et al., 2019*) suggest the possibility of neurodegeneration. Moreover, patients with WS may exhibit hypogonadism of the testes in males and ovaries in females and, thus, reduced fertility (*Huang et al., 2006*). To assess whether either nematode ortholog (CeWRN-1 or MUT-7) is required for fertility, the number of eggs produced during the first 3 days of adulthood was counted. We found that loss of CeWRN-1 reduced the total brood size from an average of 213 in the wild type to 185 eggs in the *Cewrn-1(gk99)* mutant (*Figure 5—figure supplement 1*). Loss of MUT-7 exonuclease activity greatly reduced brood

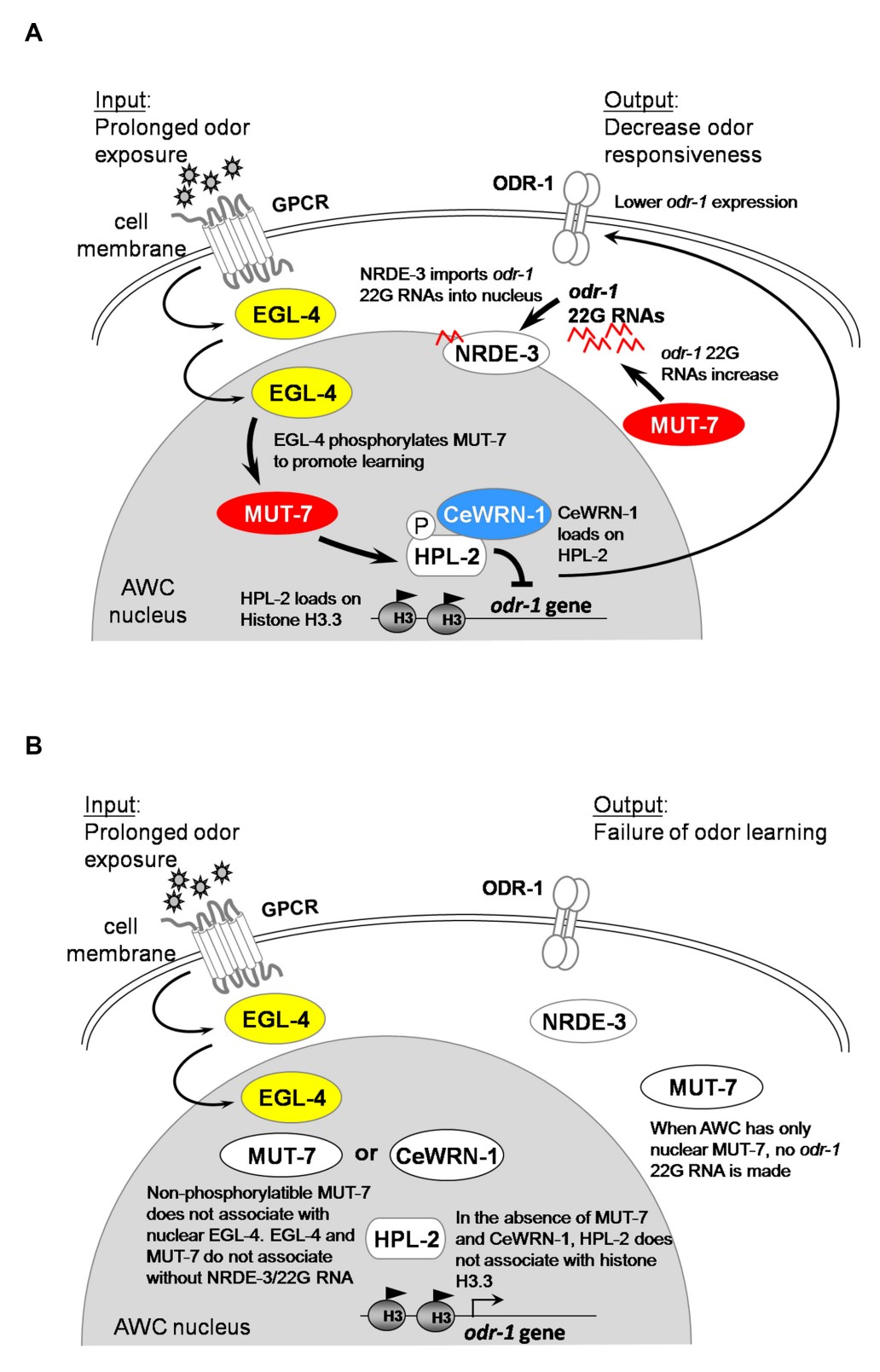

**Figure 5.** Models of the involvement of MUT-7 and CeWRN-1 in long-term olfactory learning in AWC. (**A**) In wild-type AWC neurons, prolonged odor exposure causes EGL-4 to enter the nucleus. *odr-1* 22G RNAs are generated by cytoplasmic MUT-7 and shuttled into the nucleus by NRDE-3. Once the small RNA enters the nucleus, EGL-4 phosphorylates nuclear MUT-7, which directs the association between CeWRN-1 and HPL-2. HPL-2 then associates with methylated H3.3, thereby downregulating *odr-1* transcription. Lower levels of the ODR-1 protein are predicted to promote olfactory learning. (**B**)
*Figure 5 continued on next page*

Figure 5 continued

Loss of MUT-7 from the cytoplasm inhibits *odr-1* 22G RNA production. Although EGL-4 accumulates in the nucleus, the lack of induced small RNA import leads to the disruption of the MUT-7 and EGL-4 interaction, the MUT-7 and CeWRN-1 interaction, and the CeWRN-1 and HPL-2 interaction. The lack of these associations affects the *odr-1* mRNA-downregulating signal because of the failure of HPL-2 loading on histones. The continued expression of ODR-1 results in defects in olfactory learning. Supplemental Information titles and legends.

The online version of this article includes the following figure supplement(s) for figure 5:

**Figure supplement 1.** Loss of MUT-7 significantly decreases brood size.
**Figure supplement 2.** Loss of MUT-7 significantly affects germline cell death.
**Figure supplement 3.** Images of cell corpses in wild-type and *mut-7(pk204)* animals.

size to an average of 96 eggs, and *mut-7;Cewrn-1* double-mutant animals showed an average of only 68 eggs per brood. We also asked whether the reduced brood size was the result of germline cell death. The wild-type and *Cewrn-1* mutant animals showed similar numbers of cell corpses per gonad arm, while the *mut-7* mutants showed a larger number of cell corpses (*Figure 5—figure supplements 2–3*). Thus, the MUT-7 exonuclease is required for germ cell viability, but the CeWRN-1 helicase may not be sufficient. The findings obtained in *C. elegans* may provide an opportunity to study how the exonuclease domain of the human WRN protein is involved in reproductive health in the future.

The obtained molecular evidence of the WRN homolog's $3'-5'$ exonuclease function indicates that the residues critical for $3'-5'$ exonuclease catalytic activity are conserved between MUT-7 and the exonuclease domain of mammalian WRN (*Ketting et al., 1999*). The human WRN protein exhibits both DNase and RNase activities (*Suzuki et al., 1999*). The analysis of the crystal structures of the WRN exonuclease and helicase domains indicated that they are likely to cooperate in repairing DNA (*Perry et al., 2006*). Nematode MUT-7 is involved in germline transposon silencing via the RNA interference pathway (*Ketting et al., 1999*). Loss of MUT-7 results in the loss of transposon silencing and increased activity of the DNA repair machinery in response to transposon-mediated DNA double-strand breaks in germ cells (*Wallis et al., 2019*). The location in the germline where this transposition occurs (*Wallis et al., 2019*) is consistent with the gonad area in which we observed cell corpses in *mut-7* mutants (*Figure 5—figure supplement 3*). Furthermore, the RNAi pathway silences transposon activity by producing abundant 22G RNAs in the nematode germline that serve to repress transposase expression (*Gu et al., 2009*; *Halic and Moazed, 2009*). Our work indicates that the prolonged external stimulation of *C. elegans* activates 22G RNA synthesis via the function of MUT-7 in the cytoplasm. Moreover, the entry of 22G RNA into the nucleus guides the interaction between CeWRN-1 and HPL-2 to form repressive chromatin. Small RNAs involved in WS process have also been observed in WS mouse models, in which some microRNAs, such as miR124, show differential expression compared to that in wild-type mice (*Dallaire et al., 2012*). Although evidence of the importance of siRNA in WS is currently lacking, we systematically demonstrated in the present study that the synthesis and shuttling of siRNA into the AWC nucleus are required for CeWRN-1-dependent olfactory learning. The identity of all siRNAs involved in plasticity is unknown, as is the full catalog of histone modifications that dynamically regulate neuronal plasticity. Indeed, a few recent reports indicate that human WS may result not from the loss of genome integrity but from changes in epigenetic marks such as DNA methylation patterns (*Maierhofer et al., 2019*). Thus, the identification of molecular mechanisms that link environmental challenges to epigenetic changes in well-defined models may provide an opportunity to elucidate the pathogenesis of segmental progeroid syndromes. For example, this work sets up the testable hypothesis that the exonuclease domain of human WRN may be involved in the regulation of small RNAs to protect the genome from changes that promote aging.

Overall, our study reveals two different cellular roles of the MUT-7 $3'-5'$ exonuclease: amplification of endogenous *odr-1* 22G RNAs in the cytoplasm and mediation of the association between MUT-7 and EGL-4 in the nucleus. We present evidence that the NRDE-3-mediated shuttling of small RNA into the nucleus is required for MUT-7 to associate with EGL-4. Our olfactory behavioral analysis shows that the MUT-7 exonuclease and CeWRN-1 helicase not only function in the same pathway to promote learning but also associate with each other in the nucleus after odor training. Finally, we show that the CeWRN-1 helicase interacts with the heterochromatin-binding protein HPL-2 to mediate the introduction of repressive chromatin marks on histone H3, which are required to

downregulate ODR-1 expression (*Figure 5*). Further studies on the different roles of WRN exonuclease and helicase in regulating behavioral plasticity may not only reveal the pathogenic mechanism of WS but also contribute to the development of new molecular therapeutic strategies.

# Materials and methods

## Key resources table

| Reagent type (species) or resource | Designation | Source or reference | Identifiers | Additional information |
|---|---|---|---|---|
| Strain, strain background (*Caenorhabditis elegans*) | N2 | Caenorhabditis Genetics Center | RRID:WB-STRAIN: WBStrain00000001 | Genotype: wild type |
| Strain, strain background (*C. elegans*) | *Cewrn-1(gk99)* | Caenorhabditis Genetics Center | RRID:WB-STRAIN: WBStrain 00035559 | Genotype: wrn-1 (gk99) II |
| Strain, strain background (*C. elegans*) | *hpl-2(tm1489)* | Caenorhabditis Genetics Center | RRID:WB-STRAIN: WBStrain00030670 | Genotype: hpl-2 (tm1489) III |
| Strain, strain background (*C. elegans*) | *mut-7(ne4255)* | Caenorhabditis Genetics Center | RRID:WB-STRAIN:WB Strain00040468 | Genotype: mut-7(ne4255) III |
| Strain, strain background (*C. elegans*) | *mut-7(pk204)* | Caenorhabditis Genetics Center | RRID:WB-STRAIN: WBStrain00028942 | Genotype: mut-7(pk204) III |
| Strain, strain background (*C. elegans*) | *nrde-3(gg66)* | Caenorhabditis Genetics Center | RRID:WB-STRAIN:WB Strain00040756 | Genotype: nrde-3(gg66) X |
| Recombinant DNA reagent | p*ceh-36^{prom3}*-pPD95.75 vector | Oliver Hobert Lab | | AWC-specific promoter *ceh-36^{prom3}* |
| Recombinant DNA reagent | p*Cewrn-1::GFP* | This study | Bi-Tzen Juang Lab | 2.5 kb upstream of the *Cewrn-1* start site of translation |
| Recombinant DNA reagent | p*AWC::GFP::MUT-7* | Noelle D. L'Etoile Lab *Juang et al., 2013* | | |
| Recombinant DNA reagent | p*AWC::4XNLS::mCherry::MUT-7* | This study | Bi-Tzen Juang Lab | 4XNLS::mCherry obtained from pGC302 (Addgene) |
| Recombinant DNA reagent | p*AWC::4XNLS::GFP::MUT-7* | This study | Bi-Tzen Juang Lab | 4XNLS::GFP obtained from pGC240 (Addgene) |
| Recombinant DNA reagent | p*AWC::MUT-7::GFP* | This study | Bi-Tzen Juang Lab | MUT-7 was amplified from *yk443* plasmid |
| Recombinant DNA reagent | p*AWC::CeWRN-1::GFP* | This study | Bi-Tzen Juang Lab | *Cewrn-1* cDNA from *yk1276* plasmid |
| Recombinant DNA reagent | *ODR-1::GFP* | This study | Noelle D. L'Etoile Lab | AID::3xFLAG, 1 kb downstream of *odr-1.b* stop codon |

*Continued on next page*

*Continued*

| Reagent type (species) or resource | Designation | Source or reference | Identifiers | Additional information |
|---|---|---|---|---|
| Recombinant DNA reagent | pAWC::GFP::MUT-7(W812Amber) | This study | Bi-Tzen Juang Lab | Site-directed mutagenesis reaction Technologies |
| Recombinant DNA reagent | podr-3::NLS::VC155::EGL-4 | This study | Bi-Tzen Juang Lab | Add NLS sequence and BiFC analysis |
| Recombinant DNA reagent | podr-3::VC155::EGL-4. | This study | Bi-Tzen Juang Lab | BiFC analysis |
| Recombinant DNA reagent | pAWC::VN173::MUT-7 | This study | Bi-Tzen Juang Lab | BiFC analysis |
| Recombinant DNA reagent | pAWC::NLS::VN173::MUT-7 | This study | Bi-Tzen Juang Lab | Add NLS sequence and BiFC analysis |
| Recombinant DNA reagent | pAWC::VN173::MUT-7 | This study | Bi-Tzen Juang Lab | BiFC analysis |
| Recombinant DNA reagent | pAWC::MUT-7::HA::VC155 | This study | Bi-Tzen Juang Lab | Add HA taq |
| Recombinant DNA reagent | pAWC::MUT-7::c-Myc::VN173 | This study | Bi-Tzen Juang Lab | Add c-Myc taq |
| Recombinant DNA reagent | pAWC::HPL-2::VC155 | Noelle D. L'Etoile Lab | | *Juang et al., 2013* |
| Recombinant DNA reagent | pAWC::HIS-71::VN173 | Noelle D. L'Etoile Lab | | *Juang et al., 2013* |
| Recombinant DNA reagent | pAWC::CeWRN-1::VN173 | This study | Bi-Tzen Juang Lab | BiFC analysis |
| Commercial kit | QuikChange Lightning Site-Directed Mutagenesis Kit | Agilent Technologies | Agilent:210518 | |
| Commercial kit | TURBO DNA-free Kit | Thermo Fisher: Invitrogen | Catalog number::AM1907 | |
| Commercial kit | Multiscribe Reverse Transcriptase | Thermo Fisher: Applied Biosystems | Catalog number: 4311235 | |
| Commercial kit | TaqMan Universal PCR Master Mix | Thermo Fisher: Applied Biosystems | Catalog number: 4326708 | |
| Commercial kit | iScrpt cDNA Synthesis Kit | Bio-Rad | Catalog number: 1708890 | |
| Commercial kit | iTaq Universal SYBR Green Supermix | Bio-Rad | Catalog number: 1725120 | |
| Software, algorithm | GraphPad | Prism | GraphPad Prism 8.0.1 | |
| Software, algorithm | MetaMorph | Molecular Devices | Version 7.8.13.0 | |

## Worm strains

Bristol N2 was used as the wild-type strain: *Cewrn-1(gk99)*; *hpl-2(tm1489)*; *mut-7(ne4255)* and *mut-7 (pk204)*; *nrde-3(gg66)*. Nematode strains used in this work were provided by the CGC, which is funded by NIH Office of Research Infrastructure Programs (P40 OD010440). Strains were maintained at 20°C on NGM plates seeded with *Escherichia coli* OP50.

## Plasmid construction and transgenic strains

Almost constructs were created based on the original plasmid p*ceh-36^prom3*-pPD95.75 (a kind gift from Oliver Hobert [*Etchberger et al., 2007*]), containing an AWC-specific promoter *ceh-36^prom3*, referred to as p*AWC*. Transgenic lines were generated by injecting 20–25 ng/µl of plasmids except

BiFC constructs injected 5 ng/µl. Twenty-five nanograms per microliter of *ofm-1*::GFP and p*AWC*::mcherry was used as coinjection markers.

## Cell expression analysis of CeWRN-1

### *p*Cewrn-1::GFP

2.5 kb upstream of the *Cewrn-1* start site of translation was amplified and placed into pPD95.75 containing a GFP on the plasmid backbone. This plasmid was used in *Figure 4—figure supplement 2*.

## AWC-specific expression of MUT-7 and CeWRN-1

### p*AWC*::GFP::MUT-7

This construct was created in *Juang et al., 2013*. After microinjection into N2 animals, the transgenic line was exposed to TMP and UV for integration purpose. The integrants were selected in the F2 generation by observing 100% transmission of the coinjection markers. To clean the genome of TMP-induced mutations, this strain was outcrossed to N2 through at least three generations. This strain was then crossed to *mut-7(pk204)* background. This plasmid was used in *Figure 2*.

### p*AWC*::4XNLS::mCherry::MUT-7

4XNLS::mCherry obtained from pGC302 (Addgene) was PCR amplified with primers that contained *BamHI* and *XmaI* sites and inserted into p*ceh-36*^prom3^-pPD95.75 pre-cut with the same restriction enzymes. The *mut-7* cDNA was amplified from p*AWC*::GFP::MUT-7 containing a *XmaI* and a *KpnI* restriction enzyme site. The PCR product and the previous p*AWC*::4XNLS::mCherry plasmid were digested with *XmaI* and *KpnI* and ligated together. This plasmid was used in *Figure 2A*.

### p*AWC*::4XNLS::GFP::MUT-7

4XNLS::GFP obtained from pGC240 (Addgene) was PCR amplified and contained XmaI at the 5′ end. The product was digested with *XmaI* and *XhoI* within the GFP and inserted into p*AWC*::GFP::MUT-7 pre-cut with the same restriction enzymes. This plasmid was used in *Figure 2B,C*.

### p*AWC*::MUT-7::GFP

The full-length cDNA encoding MUT-7 was amplified from *yk443* containing *KpnI* restriction enzyme site at both ends and in-frame inserted into the p*ceh-36*^prom3^-pPD95.75. This plasmid was used in *Figure 2*.

### p*AWC*::CeWRN-1::GFP

The *Cewrn-1* full-length cDNA from *yk1276* kindly provided by Yuji Kohara was amplified by PCR and contained *KpnI* at the 3′ end. The product was digested with *KpnI* and ligated into the p*ceh-36*^prom3^-pPD95.75 cut with *SmaI* and *KpnI*. This plasmid was used in *Figure 4B*.

### ODR-1::GFP

To tag *odr-1* locus using CRISPR/Cas9, the repair template construct (pNLZ21) was made by Gibson assembly combining the following components: 1.1 kb upstream of *odr-1.b* stop codon, N-terminal mEGFP added with Intron-embedded LoxP-flanked (Myc, let-858 Terminator, Cbr-unc-119, and reverse p*hsp16-41*::CRE:tbb-2 3′UTR) with C-terminal mEGFP, AID::3xFLAG, 1 kb downstream of *odr-1.b* stop codon, and pUC-118. mEGFP was derived from pMB66 (Addgene plasmid #19329). Cbr-unc-119 was derived from pCFJ150 vector (Addgene plasmid #19329). let-858 Terminator, p*hsp16-41*::CRE::tbb-2 3′UTR, and AID::3xFLAG were derived from pJW1584 (Addgene plasmid #121055). pU6:sgRNA (F+E) (pNLZ22) targeting *odr-1* with guide sequence 5′-ggcgtcataggcgg-taacgg was derived from pDD162 (Addgene plasmid #46149). To generate JZ2147:*odr-1(py7)* (*odr-1*::mEGFP::AID::3xFLAG), pJW1259 (p*eft-3*::Cas9::tbb-2 3′UTR) (Addgene plasmid #61251), the repair template, and pU6::sgRNA targeting *odr-1* were injected into *unc-119(ed3)* worms with coinjection markers. Progeny that moved like wild type but without fluorescent coinjection markers were PCR examined and singled out. To excise LoxP-flanked cassette, pDD104 (p*eft-3*::CRE::tbb-2 3′UTR) (Addgene plasmid #47551) was injected into verified *odr-1*-tagged worms. Unc progeny was singled out and verified by PCR. Then, *unc-119* worms carrying the *odr-1*::mEGFP fragment were crossed with wild type to remove *unc-119(ed3)* allele. This plasmid and strain were used in *Figure 2D*.

## Site-directed mutagenesis in phosphorylation sites of MUT-7

All constructs created by site-directed mutagenesis were from our previous work (*Juang et al., 2013*) except p*AWC*::GFP::MUT-7(W812Amber). The primers used for creating W812 Amber substitution were 5′-CCACTGGAAGAATGGTAGAATCGTATGCTTCATATC and 5′-GATATGAAGCATACGATTCTACCATTCTTCCAGTGG, and the site-directed mutagenesis reaction was performed by the QuikChange Lightning Site-Directed Mutagenesis Kit (Agilent Technologies). These plasmids were used in *Figure 3A*.

## BiFC analysis

Two *C. elegans* BiFC plasmids, pCE-BiFC-VN173 and pCE-BiFC-VC155, were obtained from AddGene. All site-directed mutagenesis reactions were performed by QuikChange Lightning Site-Directed Mutagenesis Kit (Agilent, 210518). These plasmids were used in *Figures 3C* and *4A*.

a.  p*odr-3::NLS::VC155::EGL-4*. This construct used was made by replacing the GFP fragment of p*odr-3*::NLS::GFP::EGL-4 (*Lee et al., 2010b*) with a VC155 fragment that was amplified from pCE-BiFC-VC155 with two primers containing a *BamHI* site at the 5′ end of the forward and reverse primers (5′-GGATCCGACAAGCAGAAGAACGGCAT and 5′-GGATCCCTTGTACAGCTCGTCCATG). The resulting plasmid was sequenced to confirm the correct orientation.

b.  p*odr-3::VC155::EGL-4*. This construct was created by two rounds of site-directed mutagenesis on p*odr-3*::NLS::VC155::EGL-4. First, a start codon ATG was in frame inserted into upstream of the VC155 with two primers designed by primerX software (http://www.bioinformatics.org/primerx/cgi-bin/DNA_1.cgi) (5′-CGTAAGGTAGGATCCATGGACAAGCAGAAGAAC and 5′-GTTCTTCTGCTTGTCCATGGATCCTACCTTACG). Next, the start codon ATG of NLS was disrupted with two primers (5′- GAAAATCAACTGGAAATAAGCCCAAAGAAGAAGCG and 5′-CGCTTCTTCTTTGGGCTTATTTCCAGTTGATTTTC).

c.  p*AWC::VN173::MUT-7*. A VN173 fragment was PCR amplified using pCE-BiFC-VN173 as template with primers that contained *BamHI* at the both ends and cloned into the p*ceh-36^{prom3}*-pPD95.75 plasmid. The *mut-7* cDNA was PCR amplified with two primers that contained *KpnI* and *EcoRI* (5′-tttgcgGGTACCATGGAAGAAGAACCGTACAAAAGA and 5′-CAGTTGGAATTCTCAACATTCCTGGCTGGTG) and cloned into the previous plasmid pre-cut with the same enzymes.

d.  p*AWC::NLS::VN173::MUT-7*. A 4XNLS fragment was PCR amplified from pGC240 (AddGene) and contained *BamHI* at both ends and inserted into the p*AWC*::VN173::MUT-7 with the same restriction enzyme.

e.  p*AWC::MUT-7::HA::VC155*. The AWC-specific promoter, *ceh-36^{prom3}*, was fused upstream of the *mut-7* cDNA and cloned into pCE-BiFC-VC155 using the restriction enzymes *XmaI* and *SacII*. To analyze whether this version of MUT-7 functions normally, p*AWC*::MUT-7::HA::GFP was made by replacing the VC155 fragment with GFP using the restriction enzymes *KpnI* and *NotI*. This plasmid was used in *Figure 4—figure supplement 3*.

f.  p*AWC::MUT-7::c-Myc::VN173*. The AWC-specific *ceh-36^{prom3}* promoter was fused upstream of the *mut-7* cDNA and cloned into pCE-BiFC-VN173 using the restriction enzymes *SphI* and *SacII*.

g.  p*AWC::HPL-2::VC155* and p*AWC::HIS-71::VN173*. These constructs were created in *Juang et al., 2013*.

h.  p*AWC::CeWRN-1::VN173*. This construct was made by replacing the *his-71* genomic DNA of p*AWC*::HIS-71::VN173 with the *Cewrn-1* cDNA which was amplified with two primers containing *NheI* and *KpnI* (5′-cttgGCTAGCATGATAAGTGATGATGACGATCTACC and 5′-ctttGGTACC AAGTTTGAATTTCTTCAATGGAGG).

## Behavioral assay

Olfactory behavioral assay was performed as described previously (*Colbert and Bargmann, 1995*). Briefly, ~200 adult worms that were grown on HB101-seeded plates for 5 days at 20℃ were collected and split into two 1.5 ml microcentrifuge tubes. To remove bacterial contamination, animals were washed three times with S-basal buffer. For odor training, animals were pre-exposed to 1.5 ml of diluted buffer (10 μl of butanone per 100 ml S-basal buffer), while a control population was pre-exposed to S-basal buffer only. After 80 min, animals were washed twice with S-basal buffer and once with water to get rid of butanone. Animals were then placed on 10 ml of chemotaxis agar (1.6% agar in 5 mM potassium phosphate [pH 6.0], 1 mM CaCl$_2$, and 1 mM MgSO$_4$) in a 9 cm

diameter Petri dish. One microliter drop of butanone source diluted 1:1000 in ethanol and 1 µl drop of ethanol source were spotted on each side of the plate with 1 µl of 1 M sodium azide at the same spots at the beginning of the assay. After 2 hr, worms were scored for CI index calculation: the number of animals roaming near the attractive odor source minus the number of animals roaming near the control ethanol source and then the difference was divided by the total animals on the assay plate. Animals were kept at 20℃ through all the assay steps.

## BiFC assay

Transgenic worms expressing a pair of BiFC constructs were picked at L4 stage and incubated on HB101-seeded plate for 5 days at 20℃. Young adult animals were washed with S-basal buffer for three times and then spilt into two tubes: one incubates in S-basal buffer (naïve worm); another pre-exposes to butanone-diluted buffer (odor-trained worms). The tubes were rotated for 80 min at 20℃. Twenty to 30 worms were mounted on an agarose pad with adding 1 µM of NaN$_3$, and then took pictures at the same illumination and exposure using an Upright Microscope (Leica DM6 B) at 63× magnification. All tested worms are analyzed by taking continuous Z-section fluorescence images at 0.3 µm intervals throughout the thickness of the pharynx. We analyze carefully all the images to decide whether BiFC signals are observed or not. The quantitation of BiFC is determined by dividing the number of worms showing BiFC signals by the total number of worms tested.

## Brood size assay

To measure the brood size of worms, single L4 worm was picked and grown on a NGM plate seeded with *E. coli* OP50 at 20℃. Once reached adulthood, the animal was gently transferred to a new plate, and its eggs laid on the original plate were scored. This step was repeated every 24 hr until Day 3.

## Gonad cell death assay

Five L4 worms were placed on an OP50-seeded plate and incubated at 20℃. Once animals had grown to adulthood, cell corpses in the gonad were counted at 12, 24, 48, and 96 hr in DIC microscopy.

## Quantitative real-time PCR

Quantitative real-time PCR was performed as described previously (*Juang et al., 2013*). Briefly, adult animals from three bacterial seeded plates were collected and washed with S-basal buffer. One hundred to 200 animals were applied for a behavioral assay, and the remaining animals were isolated total RNA by using TRIZOL extraction (*Chomczynski and Sacchi, 1987*). RNA was purified by 1-bromo-3-chloropropane, precipitated by isopropanol, and resuspended in RNAase-free water. The genomic DNA in the RNA samples was removed by using TURBO DNA-free Kit (Invitrogen).

### Quantitation of odr-1 22G RNA

The total RNA from entire worms was used in measuring 22G RNA quantitation in *Figure 2C*. All *odr-1* 22G RNAs were originally provided from *Gu et al., 2009*, and the most abundant species termed *odr-1.7* (GCAAACATATTGAGGGTAAGT) was used to design Taqman probe and primers for quantization of 22G RNA (*Juang et al., 2013*). Forty-eight nanogram of total RNA was applied for cDNA synthesis by Multiscribe Reverse Transcriptase (Applied Biosystems). Quantitative real-time PCR was prepared by mixing cDNA, fluorogenic probe, and TaqMan Universal PCR Master Mix (Applied Biosystems) in triplicate for each sample. Thermocycling conditions carried out on a Roche LC-480 II instrument were denaturation at 95℃ for 10 min, followed by 50 cycles of 15 s at 95℃ and 1 min at 60℃. The threshold cycle number of log-based fluorescence (Ct) was obtained, and the relative expression level (dCt) of *odr-1* 22G RNA was normalized to a mature small nuclear RNA control (sn2343, Applied Biosystems). The fold change of 22G RNA expression was calculated by the ratio of odor-trained over naive populations.

### Quantitation of *odr-1* mRNA

One milligram of total RNA was applied for cDNA synthesis by using iScrpt cDNA Synthesis Kit (Bio-Rad). Twenty microliters of dye-based quantitative PCR was prepared by adding 500 nM of forward

and reverse primers, 2 µl of cDNA, and 10 µl of 2× iTaq Universal SYBR Green Supermix (Bio-Rad). The cycling program was run 50 cycles and each cycle included 95℃ for 15 s and 60℃ for 1 min. The primers for *odr-1* were 5′-gcgaagacccctaccattta and 5′-cgctggcaacatttcattta, and the primers for internal control *act-3* were 5′-cggtatgggacagaaggac and 5′-ggaagcgtagagggagagga. The mRNA expression level of *odr-1* was determined by the Ct value and then normalized to *act-3*. The fold change of odr-1 mRNA by prolonged odor exposure was measured by the ratio of odor-trained over naïve worms and shown in *Figure 2—figure supplement 2*.

## Acknowledgements

We thank Paul J Hagerman for critical reading and helpful discussion. We thank Jung-Hsiang Chen, Chin-Fu Wang, and Jay Shieh (Department of Materials Science and Engineering, National Taiwan University); Yan-Hwa Wu Lee, Chih-Sheng Lin, and Yun-Ming Wang (National Chiao Tung University); and Yuh-Jyh Jong (Kaohsiung Medical University) for their help in the early development of the B-TJ laboratory. We thank Chan-Hsien Yeh, Chiao-Hui Chuang, Chun-Yen Teng, Yi-Yin Chen, and Huang-Chin Lin for figure editing and software teaching. We thank the Caenorhabditis Genetics Center and the National Bioresource Project for worm strains and Yuji Kohara for yk cDNA clones. We thank the assistance from the Taiwan *C. elegans* core facility (CECF). B-TJ acknowledges support from the Ministry of Science and Technology, Taiwan (103–2311-B-009–003-MY2 and 105–2311-B-009–002-MY3) and the higher education sprout project of the National Chiao Tung University and Ministry of Education, Taiwan. NDL acknowledges support from the National Institutes of Health (2R01DC005991 and R01DC015758).

## Additional information

### Funding

| Funder | Grant reference number | Author |
|---|---|---|
| Ministry of Science and Technology, Taiwan | 103-2311-B-009 -003-MY2 | Bi-Tzen Juang |
| Ministry of Science and Technology, Taiwan | 105-2311-B-009 -002-MY3 | Bi-Tzen Juang |
| National Institutes of Health | 2R01DC005991 | Noelle D L'Etoile |
| National Institutes of Health | R01 DC015758 | Noelle D L'Etoile |
| National Chiao Tung University | The higher education sprout project of the National Chiao Tung University and Ministry of Education, Taiwan | Bi-Tzen Juang |

The funders had no role in study design, data collection and interpretation, or the decision to submit the work for publication.

### Author contributions

Tsung-Yuan Hsu, Conceptualization, Data curation, Formal analysis, Validation, Investigation, Visualization, Methodology, Writing - original draft, Project administration, Writing - review and editing; Bo Zhang, Data curation, Validation, Methodology; Noelle D L'Etoile, Conceptualization, Data curation, Formal analysis, Supervision, Funding acquisition, Validation, Investigation, Visualization, Methodology, Writing - original draft, Writing - review and editing; Bi-Tzen Juang, Conceptualization, Data curation, Formal analysis, Supervision, Funding acquisition, Validation, Investigation, Visualization, Methodology, Writing - original draft, Project administration, Writing - review and editing

### Author ORCIDs

Tsung-Yuan Hsu https://orcid.org/0000-0001-7234-7810
Bi-Tzen Juang https://orcid.org/0000-0003-1210-0992

Decision letter and Author response
Decision letter https://doi.org/10.7554/eLife.62449.sa1
Author response https://doi.org/10.7554/eLife.62449.sa2

## Additional files

### Supplementary files

• Supplementary file 1. Asymmetric expression of *str-2* in AWC neurons. The cell fate of the AWC neuron was examined by quantifying the asymmetric expression of p*str-2*::DsRed. Animals were scored in three categories according to p*str-2*::DsRed expression either in AWC (0AWCp*str-2* ON), in only one AWC (1AWCp*str-2* ON), or in both AWCs (2AWCp*str-2* ON).

• Supplementary file 2. Nuclear accumulation of EGL-4 after prolonged butanone exposure. Integrated GFP-tagged EGL-4 (termed pyIs500) was expressed in wild-type, *hpl-2(tm1489)*, and *Cewrn-1 (gk99)* animals. The nuclear expression of EGL-4 was scored in naïve and odor-trained animals.

• Transparent reporting form

### Data availability

All data generated or analysed during this study are included in the manuscript and supporting files.

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
