## [Decision Letter]

**Acceptance summary:**

Using a powerful combination of approaches including in vivo detection of protein-protein interaction, this paper examines the functions of the two domains of the human Werner syndrome protein WRN in *C. elegans* behavioral plasticity. This work shows that these two domains, encoded separately by the *C. elegans* genes mut-7 and wrn-1, have key but distinct roles in a pathway by which the nuclear import of small, non-coding 22G RNAs leads to changes in neuronal gene expression that mediate olfactory adaptation. These findings shed important light on the roles of small RNAs in neuronal plasticity as well as the functions of the two domains of WSP in regulating chromatin state and gene expression.

**Decision letter after peer review:**

Thank you for submitting your article "*C. elegans* orthologs MUT-7/CeWRN-1 of Werner syndrome protein regulate neuronal plasticity" for consideration by *eLife*. Your article has been reviewed by three peer reviewers, and the evaluation has been overseen by a Reviewing Editor and Piali Sengupta as the Senior Editor. The following individual involved in review of your submission has agreed to reveal their identity: Rachel Arey (Reviewer #1).

The reviewers have discussed the reviews with one another and the Reviewing Editor has drafted this decision to help you prepare a revised submission.

The editors have judged that your manuscript is of interest, but, as described below, extensive revisions are required before it can be published. We would like to draw your attention to changes in our revision policy that we have made in response to COVID-19 (https://elifesciences.org/articles/57162). First, because many researchers have temporarily lost access to the labs, we will give authors as much time as they need to submit revised manuscripts. We are also offering, if you choose, to post the manuscript to bioRxiv (if it is not already there) along with this decision letter and a formal designation that the manuscript is "in revision at *eLife*". Please let us know if you would like to pursue this option. (If your work is more suitable for medRxiv, you will need to post the preprint yourself, as the mechanisms for us to do so are still in development.)

Summary:

This paper investigates the roles of MUT-7 and CeWRN-1, two domains of the human Werner Syndrome Protein, in in experience-dependent neuronal plasticity in *C. elegans*. In previous work, the authors showed that small RNA-mediated downregulation of odr-1 in the AWC neuron is important for olfactory adaptation to butanone. Here, the authors present evidence, including studies using an innovative and exciting BiFC approach, that suggest that (1) MUT-7 is required in the cytoplasm for siRNA synthesis, (2) MUT-7 interacts with CeWRN-1 in the nucleus, (3) NRDE-3 translocation into nucleus promotes HPL-2 binding to chromatin at specific loci. Based on these findings, the authors present an interesting model that would contribute significantly to our understanding of how small RNA pathways target endogenous genes to regulate neuronal function. However, as presented, there are significant concerns about several aspects of the model. These need to be addressed with additional experiments (including some important missing controls) as outlined below, as well as more rigorous statistical analysis. Further, for the model to significantly advance the understanding of mechanisms underlying olfactory adaptation, additional studies need to be carried out to address the role of NRDE-3 in this process.

Essential revisions:

1) The manuscript requires extensive editing for clarity and grammar. The results should be reorganized to follow the flow of the figures. For example, the Cewrn-1 qPCR data in Figure 2C is not referred to until the end of the Results section. Further, the data presented in Figure 4 do not flow with the logic of the text, making it very difficult to follow the authors' results and conclusions. (Here, it might be useful to consider making two separate figures, one that focuses on BifC and the other on CeWRN.) There are numerous grammar issues, and the wording is sometimes awkward, making several sections of the paper very difficult to follow.

2) Statistical analyses need to be improved. There seems to be a general lack of information as to what statistical tests were applied in the figure legends. In the figure legend for 1B, the authors state "P values show T-test results by comparing the indicated odor-trained population." This suggests that multiple individual t-tests are being performed between groups, in which case it is necessary to account for multiple comparisons.

3) There are concerns about the experimental design of the BiFC experiments. In Figure 3B, the naïve and pre-exposure animals appear to be different ages. It also seems unusual that the BiFC image for the naïve animals is completely black and does not show any background fluorescence as seen in the pre-exposure image. The BiFC signals of the naïve worms should be reported as controls, since the NLS::EGL-4 is not equivalent to untagged EGL-4. The experiment to test whether MUT-7 associates with EGL-4 in the nucleus in the absence of 22Gs is not well designed – it would be better to have a wild type nuclear MUT-7 expressed in AWC in a genetic background that prohibits 22G production (mut-7 allele or otherwise). Using W812Amber only indicates that this version of the protein does not associate with EGL-4. Further, Figure 2A uses NLS-mCherry-MUT-7, but the Results section and Figure 2B use NLS-GFP-MUT-7. Why are different fluorescent proteins used here? The mCherry-NUT-7 (no NLS) needs to be tested for its ability to rescue mut-7(-). Otherwise, the defect of NLS-mCherry-MUT-7 can also be contributed by mCherry tagging, in addition to NLS.

4) BiFC data interpretation. The analyses here use a binary classification (+ or – BiFC signal). Please describe the sensitivity of the assay and how such classification is determined. For example, what is the cut off? Also, if there is no detectable signal in naïve worms and strong signal in adapted worms, does that mean the observed BiFC signal is entirely contributed by olfactory adaptation and the interactions of MUT-7-EGL-4, H3-HPL-2, CeWRN-1-MUT-7 are extremely rare in the AWC cells of naïve worms? In a revised manuscript, please address these issues in the relevant section(s).

5) Further studies are needed to strengthen the model and provide a sufficient level of new mechanistic insight. In the Results section, it is stated that "phosphorylated MUT-7 directs the heterochromatin complex to genetic loci perhaps using 22G RNA as a guide." However, since MUT-7 has not been shown to have Argonaute activity, it seems more likely that NRDE-3 would be directing a complex that contains MUT-7, CeWRN-1, and HPL-2 to target loci. Experiments to address this can be carried out using the BiFC strains already on hand. Specific questions that should be addressed are: Are 22Gs required for MUT-7 phosphorylation? How is NRDE-3 involved? The paper currently shows that NRDE-3 is required for MUT-7 and EGL-4 interaction, but what about CeWRN-1 and HPL-2 interaction, or MUT-7 and CeWRN-1 interaction? Are MUT-7, CeWRN-1, and HPL-2 in a complex together? Does NRDE-3 direct their localization? Is it possible to IP HPL-2 and probe for MUT-7 association by Western blot (or vice versa)?

6) CeWRN-1 and 22G synthesis. The reported involvement of CeWRN-1 in 22G synthesis is quite interesting. However, how CeWRN-1 may be involved in this process while localizing to the nucleus is not resolved. Because only one 22G RNA (odr-1.7) is measured, it is not clear how representative it is for the overall change of odr-1 22G RNAs. The biological significance for 22G RNA difference b/t WT transgene(-) and mut-7 transgene (-) is questionable. There are two large outliers for WT and two small outliers for mut-7. If these outliers are removed from analysis, the difference is likely to disappear. To address these concerns, we ask you to consider sRNA-seq, which is much more sensitive and comprehensive in measuring sRNA levels. If that isn't feasible, please measure the levels of at least two other odr-1 22G RNAs. Further, your paper states that "CeWRN-1 associates with the chromatin binding protein HPL-2 to promote the heterochromatin formation for silencing ODR-1 expression," but this hypothesis is not directly tested. CeWRN also binds to MUT-7, and acts in the same pathway to regulate olfactory adaptation, so what is the mechanism? This isn't tied together well. Your paper also states that the CeWRN-1 expression pattern has not been characterized, but protein localization of CeWRN-1 using immunofluorescence was performed by Lee et al., 2004. Perhaps you are referring here to particular neurons? While additional experiments are not necessary here, these issues should be addressed by revising the text.

7) Issues concerning the two mut-7 alleles used. Your paper reports that two different alleles of mut-7 differ in the butanone chemotaxis phenotype. Your subsequent experiments use the pk204 allele, which has normal butanone chemotaxis but fail to ignore the butanone after it is paired with starvation (adaptation defective). The other allele, ne4255, is defective in butanone chemotaxis to begin with. While we understand your focus on adaptation aspect, the phenotype of butanone chemotaxis is also highly relevant. Therefore, some additional experiments are needed for ne4255. At a minimum, please measure the mRNA and sRNA levels of odr-1 in ne4255, which could provide important insight necessary to build a more complete model.

8) The section is titled: "Phosphorylated MUT-7 associates with EGL-4 in the nucleus of odor-trained animals." For this to be valid, evidence needs to be presented that MUT-7 is indeed phosphorylated. Otherwise, the conclusion as currently shown in the section title needs to be revised.

[Editors' note: further revisions were suggested prior to acceptance, as described below.]

Thank you for resubmitting your work entitled "*C. elegans* orthologs MUT-7/CeWRN-1 of Werner syndrome protein regulate neuronal plasticity" for further consideration by *eLife*. Your revised article has been evaluated by Piali Sengupta (Senior Editor) and a Reviewing Editor.

The manuscript has been significantly improved but, as noted by both reviewers 1 and 2, several of the required revisions from the initial review were not adequately addressed. Before the paper can be accepted, all of these points must be addressed. Specifically, please respond to the two issues raised by reviewer 1 and the five points raised by reviewer 2. To some extent, the concerns of these two reviewers overlap. Most of these can be addressed by modifying the text of paper to tone down conclusions, provide more context, or mention alternative explanations. However, before the paper can be accepted, please also add the missing controls mentioned in reviewer 2's point #2 (which is also relevant to reviewer 1's point #1).

Reviewer #1:

In this study by Hsu et al., the authors use *C. elegans* to investigate the roles of two different domains of the human Werner Syndrome Protein (WSP) in neuronal plasticity, by taking advantage of the fact that there are 2 worm orthologs (MUT-7 and CeWRN-1) of WSP that each only contain one of the domains of interest. This paper cleverly uses these two proteins to investigate the role of different WSP domains in the regulation of neuronal plasticity, as measured by molecular and behavioral responses to olfactory conditioning.

They find that both MUT-7 (which contains a 3'-5' exonuclease domain) and CeWRN-1 (which contains helicase domains) are necessary for normal olfactory adaptation, and function in the same genetic pathway. The also find that MUT-7 is required in both the nucleus and cytoplasm of the AWC sensory neurons for proper olfactory adaptation, while CeWRN-1 is localized to the AWC nucleus. Cytoplasmic MUT-7 appears to regulate the generation of odr-1 22G RNAs, which regulate levels of the ODR-1 guanylyl cyclase, which has been previously shown to be important for olfactory adaptation.

To further study nuclear MUT-7 and CeWRN-1, the authors used BiFC, a split fluorescent protein method that uses fluorescence as an indirect measure of in vivo protein-protein interactions. They first examine interactions between MUT-7 and the PKG EGL-4, which was previously shown to act in the nucleus to regulate olfactory adaptation and genetically interacts with MUT-7. Protein-protein interactions between MUT-7 and EGL-4 were detected and appeared to be PKG phosphorylation site-dependent, and mutating putative EGL-4 phosphorylation sites on MUT-7 also disrupted behavior.

In a series of BiFC experiments carried out in a systematic manner, the authors determine that MUT-7 interacts with both EGL-4 and CeWRN-1 in the nucleus, and CeWRN-1 and EGL-4 do not directly interact. CeWRN-1 interacts with the heterchromatin promoting protein HPL-2 and is necessary for odor-training increases in HPL-2 association with histones, suggesting that it promotes heterochromatin formation and gene silencing, which was previously shown to be important for olfactory adaptation. Lastly, the interactions between MUT-7 and CeWRN-1, MUT-7 and CeWRN-1,and CeWRN-1 and HPL-2 all appear to depend on the argonaute protein NRDE-3. These data provide new insight into mechanisms regulation behavioral plasticity, and suggest that the regulation of small RNAs and chromatin by human WSP may be underappreciated causes of disease.

The conclusions of this paper are mostly well supported by the data, but there are some weaknesses.

1) The BiFC method is deployed cleverly here, but can only tell if two things are interacting at a time. The authors mitigate this disadvantage by systematically testing different interactors (give examples) and also testing how BiFC signals change in the context of mutation. However, this method cannot determine whether multiple proteins are actually in a complex. Moreover, it appears that BiFC is an all-or-none phenomenon, thus more nuanced changes in protein interactions that may subtly affect protein complex formation cannot be resolved with this method.

2) Due to the difficulty of performing biochemical analyses on proteins expressed in a single neuron, the authors are unable to directly demonstrate that MUT-7 is directly phosphorylated by the PKG EGL-4, or determine if there are changes in MUT-7 phosphorylation in response to odor training. The authors attempt to overcome these limitations by examining the behavior and protein association by BiFC of animals expressing a mutant form of MUT-7 where putative PKG/EGL-4 phosphorylation sites are mutated. These findings suggest that phosphorylation of the protein is important, but there is no direct or concrete evidence that phosphorylation is indeed occurring, as claimed in the article.

The authors addressed almost all of the previous concerns, but should still consider re-wording the sections title "EGL-4 phosphorylates MUT-7 at predicted PKG sites in the nucleus of odor-trained animals" as there is still a lack of direct evidence. Maybe something like "MUT-7 and EGL-4 interact in the nucleus of odor-trained animals in a PKG phosphorylation site-dependent manner?"

The addition of the NRDE-3 data strengthened the paper greatly, and the reorganization made the story much clearer.

Reviewer #2:

In this manuscript, Hsu et al. investigate the role of MUT-7 and CeWRN-1 in regulating neuronal plasticity in *C. elegans*. MUT-7 and CeWRN-1 are orthologs of the human Werner Syndrome protein, which has 3'-5' exonuclease and helicase domains. In previous work, the authors showed that MUT-7, HP1 homolog HPL-2, and the nuclear Argonaute NRDE-3 are required for olfactory adaptation to butanone. Naïve *C. elegans* adults typically exhibit attractive behavior towards butanone. However, adults that were pre-exposed to butanone in the absence of food no longer exhibit attractive chemotaxis towards the odorant. The authors showed previously that pre-exposure to butanone results in the small RNA-mediated downregulation of the membrane bound guanylyl cyclase gene, odr-1, in the AWC sensory neuron. In this current manuscript, Hsu et al. state the conclusions that (1) MUT-7 is required in the cytoplasm for siRNA synthesis, (2) MUT-7 interacts with CeWRN-1 in the nucleus, (3) NRDE-3 translocation into nucleus promotes HPL-2 binding to chromatin 'at specific loci'. I found that there are some intriguing observations presented in the results that warrant further investigation and could contribute to our understanding of how small RNA pathways target endogenous genes to regulate neuronal function based on environmental cues. However, there are some issues with controls for experiments performed, and, as is, the data does not significantly increase of knowledge of the mechanisms of olfactory adaptation beyond what has been shown previously.

In the revised manuscript entitled "*C. elegans* orthologs MUT-7/CeWRN-1 of Werner syndrome protein regulate neuronal plasticity", Hsu et al. address most of the concerns of the reviewers. They have added additional data to Figures 2B and 3C, included new supplemental figures, and have clarified multiple points within the text. In my opinion, these changes are satisfactory and strengthen the conclusions of the manuscript.

However, there are a few points in the decision letter that were not addressed, or were addressed in an unsatisfactory manner, that should be considered before a final decision.

1) The manuscript still contains numerous problems with grammar and clarity and needs additional editing. In addition, the flow of "Roles of nuclear and cytoplasmic MUT-7 in promoting learning" is difficult to follow and could be rewritten instead of just adding a paragraph.

2) The authors did not add the BiFC signals of naïve, control worms to Figure 3C. NLS::EGL-4 is not equivalent to EGL-4 without the nuclear localization signal, and controls should be included.

3) The reviewers' response regarding the details of the BiPAC analysis was not adequately addressed. The reviewers would like details of how quantitation was performed for determining the percentage of animals with signal. The authors say this assay exhibited "on" or "off" signal, but how was "on" determined? Was it a cutoff of pixels above background, or determined to be "on" by eye?

4) The new NRDE-3 data was not added to Figure 4 but was instead put in Supplementary file 2. This new data was also not incorporated into the model figure (Figure 5). In my opinion, the NRDE-3 data is significant for the mechanism of how MUT-7, CeWRN-1, and HPL-2 function to silence odr-1 in olfactory adaptation and should be incorporated into the main figures.

[Editors' note: further revisions were suggested prior to acceptance, as described below.]

Thank you for submitting your article "*C. elegans* orthologs MUT-7/CeWRN-1 of Werner syndrome protein regulate neuronal plasticity" for consideration by *eLife*.

The revised version of your paper has addressed nearly all of the issues raised during the previous rounds of review. However, your paper still does not provide details on the analysis and quantitation of the BiFC results. This was specifically noted by multiple reviewers in both previous rounds of review and was included in the list of required revisions. Please revise the section titled "Bimolecular fluorescence complementation (BiFC) assay" in your Materials and methods section to address this issue. Specifically, we ask that you explicitly mention that the BiFC signal is scored as an all-or-none signal by manual visual examination of multiple Z-section images. Further, please explain that the values you provide in the text and figures represent the fraction of BiFC-positive animals among the total number of animals examined.

---

## [Author Response]

Essential revisions:1) The manuscript requires extensive editing for clarity and grammar. The results should be reorganized to follow the flow of the figures. For example, the Cewrn-1 qPCR data in Figure 2C is not referred to until the end of the Results section. Further, the data presented in Figure 4 do not flow with the logic of the text, making it very difficult to follow the authors' results and conclusions. (Here, it might be useful to consider making two separate figures, one that focuses on BifC and the other on CeWRN.) There are numerous grammar issues, and the wording is sometimes awkward, making several sections of the paper very difficult to follow.

We agree the reviewer’s comments. Specifically, we rearranged Figure 4A to flow with the logic of the text. Regarding the arrangement of Figure 2C, although *Cewrn-1* qPCR was described near the end of the Results section, these data should be compared with other qPCR data sets such as *wildtype*, *mut-7*, and *mut-7* worms with different GFP tagged MUT-7 versions. Thus, for the flow of the manuscript, we feel that it is important to present *Cewrn-1* qPCR data in Figure 2C.

2) Statistical analyses need to be improved. There seems to be a general lack of information as to what statistical tests were applied in the figure legends. In the figure legend for 1B, the authors state "P values show T-test results by comparing the indicated odor-trained population." This suggests that multiple individual t-tests are being performed between groups, in which case it is necessary to account for multiple comparisons.

We use GraphPad Prism eight software to re-do the multiple comparisons in all our statistical analyses. In our figures, we indicate the p value by using two-way ANOVA analysis between the specified groups. This statistical method is now written in the figure legends.

3) There are concerns about the experimental design of the BiFC experiments. In Figure 3B, the naïve and pre-exposure animals appear to be different ages. It also seems unusual that the BiFC image for the naïve animals is completely black and does not show any background fluorescence as seen in the pre-exposure image. The BiFC signals of the naïve worms should be reported as controls, since the NLS::EGL-4 is not equivalent to untagged EGL-4.

All our BiFC experiments were performed with the same protocol as the behavioral assay except for the last step, where worms were placed on plates for chemotaxis assays. The populations were synchronized by picking L4 stage worms and examining the next generation. The animals that are imaged from young adults are as the same as age in behavioral assays. We use the same exposure time and illumination conditions in taking each image. In imaging, we focused on the regions of the worms that often had different autofluorescence. Therefore, we replace the original images with ones in which there is similar autofluorescence in both the naive and odor-trained worms in Figure 3B. We also provide all the original BiFC images from naïve and odor-trained worms in Figure 4A as Figure 4—figure supplement 3 (new Figure).

The experiment to test whether MUT-7 associates with EGL-4 in the nucleus in the absence of 22Gs is not well designed – it would be better to have a wild type nuclear MUT-7 expressed in AWC in a genetic background that prohibits 22G production (mut-7 allele or otherwise). Using W812Amber only indicates that this version of the protein does not associate with EGL-4.

To address this point, we co-expressed the N-terminal Venus tagged MUT-7 and the C-terminal Venus tagged EGL-4 in the *nrde-3*(*gg66*) mutant background which blocks *odr-1* 22Gs to shuttle into the nucleus. We did not see the BiFC signal in this case (as in Figure 4A, first low), while there are 75% odor-trained wild-type worms showing BiFC signals in Figures 3B and 3C, first row. We could not express such BiFC constructs in *mut-7*(*pk204*) mutants because the Venus tagged wild-type MUT-7 would rescue the defects of *mut-7* mutants. Further, we observed 86% wild-type worms showing BiFC signals in co-expression of NLS-EGL-4 and NLS-MUT-7 in Figure 3C last row, but only 5% of *nrde-3*(*gg66*) mutants expressing the same constructs showed BiFC signals (We added the new data to Supplementary file 2, first row, and modified the text to clarify this point). Indeed, these results indicate that the association between MUT-7 and EGL-4 in the nucleus is required in the present of 22Gs.

Further, Figure 2A uses NLS-mCherry-MUT-7, but the Results section and Figure 2B use NLS-GFP-MUT-7. Why are different fluorescent proteins used here? The mCherry-NUT-7 (no NLS) needs to be tested for its ability to rescue mut-7(-). Otherwise, the defect of NLS-mCherry-MUT-7 can also be contributed by mCherry tagging, in addition to NLS.

To address this point, we confirmed that the mCherry-MUT-7 (no NLS) rescued *mut-7(pk204)* odor learning defects (We added the new data to Figure 2B, third pair, and modified the text to clarify this point). We created in parallel the NLS-GFP-MUT-7 and NLS-mCherry-MUT-7 constructs, but the NLS-GFP-MUT-7 strain was the only one for which we were able to obtain using a standard UV/trimethylpsoralen (UV/TMP) integration method (We modified the text to clarify this point).

4) BiFC data interpretation. The analyses here use a binary classification (+ or – BiFC signal). Please describe the sensitivity of the assay and how such classification is determined. For example, what is the cut off?

The sensitivity of the BiFC assay is determined by two important controls. First, the BiFC signals are detected in butanone-trained worms rather than naïve worms in analyzing the interactions of MUT-7-EGL-4, H3-HPL-2, and CeWRN-1-MUT-7. The worms for subsequent BiFC analysis were raised in the same HB101-seeded plate as described in the Materials and methods section of the revised manuscript. Adult animals were then spilt into two tubes: one incubated in buffer (Naïve worm) and the other pre-exposed to butanone-diluted buffer (odor-trained worms). Twenty-to-thirty worms were mounted on an agarose pad with the addition of 1 μM of NaN_3_; pictures were then taken using the same illumination and exposure. A large number of odor-trained worms were observed BiFC signals compared to few naïve worms has BiFC signals (Figures 3C and 4A). Thus, we can rule out the problem of non-specific protein aggregation in the assay. The second control is that only one AWC neuron is seen the BiFC signal in each butanone-trained animal (Figures 3B, 4C and Figure 4—figure supplement 3). This reflects the fact that butanone is sensed by only one AWC (Wes and Bargmann, 2001). Indeed, we only observed BiFC signals between MUT-7-EGL-4, H3-HPL-2, and CeWRN-1-MUT-7 in one AWC neuron after butanone exposure. These results indicate that our BiFC system specifically detects protein interactions in the one butanone-sensing neuron of the worm (We modified the text to clarify this point).

Also, if there is no detectable signal in naïve worms and strong signal in adapted worms, does that mean the observed BiFC signal is entirely contributed by olfactory adaptation and the interactions of MUT-7-EGL-4, H3-HPL-2, CeWRN-1-MUT-7 are extremely rare in the AWC cells of naïve worms? In a revised manuscript, please address these issues in the relevant section(s).

The BiFC signal is binary either on or off. Thus, in a population of naïve worms few show the signal while in a population of butanone exposed ones, many showed the signal. This means that individual worms may have complexes between these proteins in the nucleus but the proportion of the population with these complexes changes with butanone treatment. This may reflect the fact that most but not all naive animals are attracted to butanone but the proportion that is attracted decreases when worms are starved in the presence of butanone.

5) Further studies are needed to strengthen the model and provide a sufficient level of new mechanistic insight. In the Results section, it is stated that "phosphorylated MUT-7 directs the heterochromatin complex to genetic loci perhaps using 22G RNA as a guide." However, since MUT-7 has not been shown to have Argonaute activity, it seems more likely that NRDE-3 would be directing a complex that contains MUT-7, CeWRN-1, and HPL-2 to target loci. Experiments to address this can be carried out using the BiFC strains already on hand. Specific questions that should be addressed are: Are 22Gs required for MUT-7 phosphorylation?

We are unable to probe directly for in vivo MUT-7 phosphorylation status as we are unable to perform immunoprecipitations and western blots for single butanone-sensing AWC neuron. We also have not been successful in isolating the full-length bacterial MUT-7 for in vitro kinase assay. Instead, since EGL-4 is a cGMP-dependent protein kinase (PKG), we predicted MUT-7’s PKG phosphorylation sites, changed these sites to alanine, and we show that the association between MUT-7 and EGL-4 depends on phosphorylation of MUT-7 (Figure 3C, third row, and Author response table 1) and also on NRDE-3 (Figure 4A first row). Thus, the 22Gs that are provided by NRDE-3 shuttling them into the nucleus are likely required for phosphorylation of MUT-7.

**Author response table 1. resptable1:** 

Strains (worms with transgenes)	BiFC signals in the nucleus of the AWC neuron (%)	
	naïve worms	odor trained worms
Wildtype ex[CeWRN-1 and MUT-7(all S/T to A)]*	3% (n=68)	4% (n=81)

* indicates wild-type worms carrying pAWC::CeWRN-1::VN173 and pAWC::VC155::MUT-7 (all S/T to A)

How is NRDE-3 involved? The paper currently shows that NRDE-3 is required for MUT-7 and EGL-4 interaction, but what about CeWRN-1 and HPL-2 interaction, or MUT-7 and CeWRN-1 interaction?

We performed the requested BiFC analysis of CeWRN-1 and MUT-7 in *nrde-3(gg99)* mutants and found that no BiFC signal was seen (We added the new data to Supplementary file 2, second row, and modified the text to clarify this point). Similarly, we found that BiFC between HPL-2 and CeWRN-1 depends on NRDE-3 (We added the new data to Supplementary file 2, third row, and modified the text to clarify this point). These new results indicate that NRDE -3 is required for both CeWRN-1 and HPL-2 interaction, and MUT-7 and CeWRN-1 interaction. This is new insight that extends the model and strengthens this paper and we are thankful for this insight by the reviewer.

Are MUT-7, CeWRN-1, and HPL-2 in a complex together?

The reviewer asks whether MUT-7, CeWRN-1, and HPL-2 form a tripartite complex. We cannot determine if MUT-7, CeWRN-1 and HPL-2 form a tripartite complex due to the following technical limitations: (1) We are unable to perform immunoprecipitations or western blots perhaps due to the fact that we are collecting material from one cell (AWC neuron) from the whole worm and we have too high background when we try; (2). BiFC reports the interaction between two candidate proteins and we have not been able to adapt it to report an interaction between three proteins. Thus, at the current stage we are limited to use BiFC to look at complexes between MUT-7, CeWRN-1, and HPL-2 and asking if NRDE-3 is required for the possibility of a tripartite complex.

Does NRDE-3 direct their localization?

Data from previous publication and this manuscript indicate that both HPL-2 and CeWRN-1 are localized within the AWC nucleus of naïve and odor adapted worms (Juang et al., 2013 and Figure 4B) and this expression is not altered in *nrde-3(gg99)* mutants (Please see Author response image 1, middle and bottom panels). Likewise, the localization of MUT-7 in both cytoplasm and nucleus does not change in *nrde-3(gg66)* mutants (please see Author response image 1, upper panel). Thus, the results suggest that NRDE-3 does not direct MUT-7, CeWRN-1 or HPL-2 localization.

**Author response image 1. sa2fig1:** 

Is it possible to IP HPL-2 and probe for MUT-7 association by Western blot (or vice versa)?

Unfortunately, neither immunoprecipitation nor western blot work when we examine proteins expressed in the single AWC (ON) cells.

6) CeWRN-1 and 22G synthesis. The reported involvement of CeWRN-1 in 22G synthesis is quite interesting. However, how CeWRN-1 may be involved in this process while localizing to the nucleus is not resolved. Because only one 22G RNA (odr-1.7) is measured, it is not clear how representative it is for the overall change of odr-1 22G RNAs.

Our previous publication has tested different *odr-1*-derived 22G RNAs and *odr-1.7* gave the most robust signals in RT-qPCR analysis for the single butanone-sensing AWC neuron (Juang et al., 2013). We found that loss of CeWRN-1 does not affect *odr-1* 22G RNA synthesis (Figure 2—figure supplement 1), but affect the increase of *odr-1* 22G RNA after prolonged butanone exposure (Figure 2C). The possibility about how CeWRN-1 may affect 22G RNA synthesis is to change histone modification via its associated protein HPL-2 and the findings will be reported in the next publication.

The biological significance for 22G RNA difference b/t WT transgene(-) and mut-7 transgene (-) is questionable. There are two large outliers for WT and two small outliers for mut-7. If these outliers are removed from analysis, the difference is likely to disappear. To address these concerns, we ask you to consider sRNA-seq, which is much more sensitive and comprehensive in measuring sRNA levels. If that isn't feasible, please measure the levels of at least two other odr-1 22G RNAs.

To address the reviewer’s concerns, we removed the two extreme outliers from the 22G RNA dataset of wild-type worms and two moderate outliers from the 22G RNA dataset of *mut-7(pk204)* mutant backgrounds. The remaining datasets are applied to GraphPad Prism software and the statistical results still show a significant difference between the two groups by using (1) two-tailed t-test (p=0.0056) and one-way ANOVA (p<0.0001).

Further, your paper states that "CeWRN-1 associates with the chromatin binding protein HPL-2 to promote the heterochromatin formation for silencing ODR-1 expression," but this hypothesis is not directly tested. CeWRN also binds to MUT-7, and acts in the same pathway to regulate olfactory adaptation, so what is the mechanism? This isn't tied together well.

This manuscript is follow-up research results to our previous publication in the journal, *Cell*, in 2013. We previously demonstrated that HPL-2 associated with *odr-1* 22G RNA in odor-trained worms by using chromatin immunoprecipitation (Juang et al., 2013). Also, *odr-1* 22G RNA associate NRDE-3 in odor-trained wild-type worms to mediate downregulation of the *odr-1* mRNA expression by using co-immunoprecipitation and RT-qPCR (Juang et al., 2013). Next, in this manuscript, we further created GFP tagged ODR-1 integrated worms by using CRISPR system to direct monitor that GFP expression is silenced in odor-trained wild-type worms (Figure 2D). Finally, we found that CeWRN-1 associated with HPL-2 in the nucleus of wild-type odor-trained worms (Figure 4C), but loss of CeWRN-1 fails to load HPL-2 on histone H3 (HIS-71) (Figure 4A, last row). Thus, our results are able to support that CeWRN-1 associates with HPL-2 to promote the heterochromatin formation for silencing ODR-1 expression. Further, the reviewer asks how CeWRN binds to MUT-7 to regulate olfactory adaptation. We provide a possibility that the nuclear MUT-7 phosphorylation is required for associate with CeWRN-1 (please see Author response table 1). This result suggests that MUT-7 phosphorylation may be an important linker to associate with CeWRN-1 in olfactory signaling.

Your paper also states that the CeWRN-1 expression pattern has not been characterized, but protein localization of CeWRN-1 using immunofluorescence was performed by Lee et al., 2004. Perhaps you are referring here to particular neurons? While additional experiments are not necessary here, these issues should be addressed by revising the text.

We agree with the reviewer that protein localization of CeWRN-1 has been shown by using in vitro immunostaining in the nuclei of germ cells, intestine cells and embryonic cells (Lee et al., 2004). In this manuscript, we performed in vivo GFP-tagged CeWRN-1 and found that CeWRN-1 expressed in the nucleus of the AWC neurons in Figure 4B.

7) Issues concerning the two mut-7 alleles used. Your paper reports that two different alleles of mut-7 differ in the butanone chemotaxis phenotype. Your subsequent experiments use the pk204 allele, which has normal butanone chemotaxis but fail to ignore the butanone after it is paired with starvation (adaptation defective). The other allele, ne4255, is defective in butanone chemotaxis to begin with. While we understand your focus on adaptation aspect, the phenotype of butanone chemotaxis is also highly relevant. Therefore, some additional experiments are needed for ne4255. At a minimum, please measure the mRNA and sRNA levels of odr-1 in ne4255, which could provide important insight necessary to build a more complete model.

Thank you for the reviewer’s comments. When we measured the level of *odr-1* 22G from *mut-7(ne4255)* mutants, we found that the *odr-1* Taqman probe failed to detect any signal in our protocol. We also tested mRNA levels in parallel and found that *odr-1* mRNA expression level was similar to *mut-7(pk204)* (T.-Y.H. and B.-T.J., unpublished data). Thus, *odr-1* transcription and mRNA accumulation are not affected by loss of MUT-7 in the *ne4255* allele and there must be another as yet discovered explanation for the defect but this is beyond the scope of this study.

8) The section is titled: "Phosphorylated MUT-7 associates with EGL-4 in the nucleus of odor-trained animals." For this to be valid, evidence needs to be presented that MUT-7 is indeed phosphorylated. Otherwise, the conclusion as currently shown in the section title needs to be revised.

As response above for comment 5, we are unable to detect directly MUT-7 phosphorylation status due to technical limitations. Instead, we found that the association between MUT-7 and EGL-4 by seeing BiFC signals (Figure 3B) is disrupted when MUT-7 PKG sites are replaced with alanine (please see Author response table 1). Therefore, as the reviewer’s suggestion, we change the title to “EGL-4 phosphorylates MUT-7 at predicted PKG sites in the nucleus of odor-trained animals”.

[Editors' note: further revisions were suggested prior to acceptance, as described below.]

Reviewer #2:[…]In the revised manuscript entitled "*C. elegans* orthologs MUT-7/CeWRN-1 of Werner syndrome protein regulate neuronal plasticity", Hsu et al. address most of the concerns of the reviewers. They have added additional data to Figures 2B and 3C, included new supplemental figures, and have clarified multiple points within the text. In my opinion, these changes are satisfactory and strengthen the conclusions of the manuscript.However, there are a few points in the decision letter that were not addressed, or were addressed in an unsatisfactory manner, that should be considered before a final decision.1) The manuscript still contains numerous problems with grammar and clarity and needs additional editing. In addition, the flow of "Roles of nuclear and cytoplasmic MUT-7 in promoting learning" is difficult to follow and could be rewritten instead of just adding a paragraph.

Thank you for the reviewer’s comment. The revised manuscript now is edited by professional editing service American Journal Experts (AJE).

2) The authors did not add the BiFC signals of naïve, control worms to Figure 3C. NLS::EGL-4 is not equivalent to EGL-4 without the nuclear localization signal, and controls should be included.

Thank you for the reviewer’s comment. The missing controls of naïve worms expressing NLS::EGL-4 with different versions of MUT-7 mutations are added in Figure 3C, from third row to last row, and Figure 4D, first row. Prolonged odor exposure to butanone causes nuclear accumulation of EGL-4 and expression of constitutively nuclear NLS::EGL-4 in the wild-type worms decreases chemotaxis toward the AWC-sensed odor, butanone, even in naïve worms (O’Halloran et al., 2009; Lee et al., 2010, Juang et al., 2013). Our previous reports indicate that nuclear EGL-4 induces adaptation of the odor-seeking behavioral response in naive worms. Indeed, the percentage of BiFC signals in naïve worms expressing NLS::EGL-4 and MUT-7 is similar to that in the odor-trained worms. We have modified the figure legends of Figures 3C and 4D to clarify this point.

3) The reviewers' response regarding the details of the BiPAC analysis was not adequately addressed. The reviewers would like details of how quantitation was performed for determining the percentage of animals with signal. The authors say this assay exhibited "on" or "off" signal, but how was "on" determined? Was it a cutoff of pixels above background, or determined to be "on" by eye?

Thank you for the reviewer’s comment. We screen the BiFC signal by using an upright microscope (Leica DM6B) at 63X magnification. All tested worms are analyzed by taking continuous Z-section fluorescence images at 0.3μm intervals throughout the thickness of the pharynx. We analyze carefully all the images to decide whether BiFC signals are observed or not. The quantitation of BiFC is determined by dividing the number of worms showing BiFC signals by the total number of worms tested.

4) The new NRDE-3 data was not added to Figure 4 but was instead put in Supplementary file 2. This new data was also not incorporated into the model figure (Figure 5). In my opinion, the NRDE-3 data is significant for the mechanism of how MUT-7, CeWRN-1, and HPL-2 function to silence odr-1 in olfactory adaptation and should be incorporated into the main figures.

We agree with the reviewer’s suggestion and the Supplementary file 2 is incorporated into the Figure 4D (new table). The NRDE-3 data was added to the figure legends of Figures 4D and 5 to clarify the mechanism of how MUT-7, CeWRN-1, and HPL-2 function to silence *odr-1* mRNA expression in olfactory adaptation.

[Editors' note: further revisions were suggested prior to acceptance, as described below.]

The revised version of your paper has addressed nearly all of the issues raised during the previous rounds of review. However, your paper still does not provide details on the analysis and quantitation of the BiFC results. This was specifically noted by multiple reviewers in both previous rounds of review and was included in the list of required revisions. Please revise the section titled "Bimolecular fluorescence complementation (BiFC) assay" in your Materials and methods section to address this issue. Specifically, we ask that you explicitly mention that the BiFC signal is scored as an all-or-none signal by manual visual examination of multiple Z-section images. Further, please explain that the values you provide in the text and figures represent the fraction of BiFC-positive animals among the total number of animals examined.

The editorial suggestions about the analysis and quantitation of the BiFC results are added to the Materials and methods section. We also explain in the text about the fraction of BiFC-positive animals among the total number of animals examined.